# Constitutive turnover of histone H2A.Z at yeast promoters requires the preinitiation complex

Michael Tramantano, Lu Sun[†], Christy Au[†], Daniel Labuz, Zhimin Liu, Mindy Chou[‡], Chen Shen[‡], Ed Luk*

Department of Biochemistry and Cell Biology, Stony Brook University, Stony Brook, United States

*For correspondence: ed.luk@stonybrook.edu

[†]These authors contributed equally to this work

Present address: [‡]Cold Spring Harbor Laboratory, Cold Spring Harbor, United States

Competing interests: The authors declare that no competing interests exist.

**Abstract** The assembly of the preinitiation complex (PIC) occurs upstream of the +1 nucleosome which, in yeast, obstructs the transcription start site and is frequently assembled with the histone variant H2A.Z. To understand the contribution of the transcription machinery in the disassembly of the +1 H2A.Z nucleosome, conditional mutants were used to block PIC assembly. A quantitative ChIP-seq approach, which allows detection of global occupancy change, was employed to measure H2A.Z occupancy. Blocking PIC assembly resulted in promoter-specific H2A.Z accumulation, indicating that the PIC is required to evict H2A.Z. By contrast, H2A.Z eviction was unaffected upon depletion of INO80, a remodeler previously reported to displace nucleosomal H2A.Z. Robust PIC-dependent H2A.Z eviction was observed at active and infrequently transcribed genes, indicating that constitutive H2A.Z turnover is a general phenomenon. Finally, sites with strong H2A.Z turnover precisely mark transcript starts, providing a new metric for identifying cryptic and alternative sites of initiation.

## Introduction

The regulation of chromatin structure and its dynamics is integral to the control of gene expression in eukaryotes. The protein core of a canonical nucleosome, modular in nature, consists of a tetramer of histones H3 and H4 [indicated as $(H3-H4)_2$] and two dimers of histones H2A and H2B (indicated as H2A-H2B) (*Arents et al., 1991*). Nucleosomal DNA, 147-bp in length, wraps around the octameric histone core in 1.7 left-handed turns making 3 minor groove contacts with each histone pair and additional contacts near the entry-exit sites with H3 (*Luger et al., 1997*). Repeating units of nucleosomes are organized along the genomic DNA in a non-random fashion with nucleosome-depleted regions (NDRs) overlapping key regulatory elements, such as promoters and replication origins (*Eaton et al., 2010*; *Yuan et al., 2005*). Nucleosome-repelling DNA sequences and ATP-dependent remodeling activities contribute to NDR formation (*Kaplan et al., 2009*; *Zhang et al., 2011*). Like a molecular sieve, chromatin blocks non-specific protein-DNA interactions and allows localized assembly of DNA binding factors, such as the general transcription factors (GTFs) and RNA polymerase (Pol) II, at the NDRs (*Rhee and Pugh, 2012*). Mutants that perturb the native nucleosome organization can lead to transcriptional derepression and initiation from aberrant start sites (*Han and Grunstein, 1988*; *Kaplan et al., 2003*; *Whitehouse et al., 2007*).

How the transcription machinery engages the nucleosome in and around a promoter and how these nucleosomes are mobilized at different stages of transcription are important questions related to the mechanism of transcriptional control. The GTFs and Pol II assemble on an 80-to-200-basepair NDR to form the 'closed' preinitiation complex (PIC) (*Rhee and Pugh, 2012*). The nucleosome immediately downstream of the NDR is termed the +1 nucleosome (*Albert et al., 2007*). In

**eLife digest** To fit the genetic information of an animal, yeast or other eukaryote into cells, DNA is tightly wound around proteins called histones to form repeating units known as nucleosomes. However, this tight winding prevents proteins from accessing the DNA, and so prevents gene transcription – the first stage of producing the molecules encoded by a gene. For transcription to take place, nucleosomes at DNA sequences called promoters must be reorganized and disassembled, thereby allowing proteins to bind to and engage these sequences and to turn nearby genes on.

H2A is a histone protein that is found in the majority of nucleosomes in yeast cells. A different form of this histone – called H2A.Z – is found in nucleosomes near the promoter of almost every gene. It is thought that nucleosomes that contain H2A.Z are recognized and disassembled as the gene turns on, but it is unclear how this happens.

To investigate how H2A.Z nucleosomes are disassembled, Tramantano et al. depleted yeast cells of various proteins thought to play a role in the disassembly process. This indicated that the proteins that transcribe genes play crucial roles in the process of disassembling the H2A.Z nucleosomes, because H2A.Z accumulated at promoters in cells that are depleted of these proteins.

Further investigation revealed that disassembled H2A.Z nucleosomes are reassembled with H2A histones, before being converted back to the H2A.Z form by an enzyme called SWR1. This turnover of H2A.Z was seen at active genes and those that are infrequently transcribed, suggesting that it is a general phenomenon.

Tramantano et al. also found that the turnover rate of H2A.Z can be used to accurately predict the sites in the DNA where transcription starts. This observation could therefore help to identify previously unknown transcription start sites. Future work could address further questions about how H2A.Z nucleosomes are disassembled. For example, what is the mechanical force that drives this process? And at what step of the transcription process does it occur?

*Saccharomyces cerevisiae*, the +1 nucleosome covers the transcription start site (TSS) of most genes. Therefore, it is expected that at some point during the transcription process the +1 nucleosome must be disassembled. It is currently unclear at what stage of transcription disassembly of the +1 nucleosome occurs. It remains possible that chromatin remodeling enzymes are required to remove the +1 nucleosome before Pol II can engage the TSS. For other promoters where the +1 nucleosome covers the TATA element and for those with an untraditional nucleosome structure called the 'fragile nucleosome' within the NDR, histone eviction likely precedes and regulates PIC assembly (*Kubik et al., 2015*; *Rhee and Pugh, 2012*). In metazoans, where the TSS is further upstream of the +1 nucleosome than in yeast (*Schones et al., 2008*), the +1 nucleosome stalls elongation (*Weber et al., 2014*). In all cases, the disassembly of these promoter-proximal nucleosomes is likely a regulatory barrier for full-length transcript synthesis (*Churchman and Weissman, 2011*; *Weber et al., 2014*).

The promoter-proximal nucleosomes at the +1 position and to a lesser extent the -1 position (upstream of the NDR) are enriched for the histone variant H2A.Z (*Albert et al., 2007*; *Raisner et al., 2005*). Together with the histone variant H3.3, H2A.Z forms nucleosomes that are labile in high salt *in vitro* (*Jin and Felsenfeld, 2007*; *Zhang et al., 2005*). In yeast, where the major H3 is similar to the H3.3 isoform in metazoans, H2A.Z is preferentially evicted from promoters during gene activation over H2A (*Santisteban et al., 2000*; *Zhang et al., 2005*; *Venters et al., 2011*). Although mutants of *HTZ1* (the gene that encodes H2A.Z in yeast) are viable and exhibited only minor defects in steady-state mRNA levels, H2A.Z is required for rapid transcriptional activation (*Dhillon et al., 2006*; *Halley et al., 2010*; *Mizuguchi et al., 2004*; *Santisteban et al., 2000*; *Zhang et al., 2005*). These findings suggest that H2A.Z nucleosomes are predisposed for disassembly to allow for a rapid transcriptional response. What drives H2A.Z nucleosome disassembly *in vivo* will be the focus of this study.

The incorporation of H2A.Z into nucleosomes is catalyzed by the ATP-dependent chromatin remodeling complex SWR1 (*Mizuguchi et al., 2004*). The ~1 megadalton SWR1 complex comprises

the catalytic core protein Swr1, a member of the Swi2/Snf2-related ATPase, plus 13 other polypeptides (*Kobor et al., 2004*; *Krogan et al., 2003*; *Mizuguchi et al., 2004*). SWR1 is targeted to promoters by its intrinsic affinity for the NDR and promoter-specific histone acetylation (*Raisner et al., 2005*; *Ranjan et al., 2013*). It catalyzes a histone replacement reaction that involves the coupled removal of a nucleosomal H2A-H2B dimer with the insertion of an H2A.Z-H2B dimer that is delivered to the enzyme by one of several histone chaperones, including Nap1, Chz1, and FACT (*Luk et al., 2007*, *2010*; *Mizuguchi et al., 2004*). The two H2A-H2B dimers in a homotypic H2A (AA) nucleosome are replaced sequentially, first generating the heterotypic H2A/H2A.Z (AZ) nucleosome as an intermediate (*Figure 1—figure supplement 1A*, step I-a) and the homotypic H2A.Z (ZZ) nucleosome as the final product (*Figure 1—figure supplement 1A*, step I-b) (*Luk et al., 2010*). The $(H3-H4)_2$ tetramer remains associated with the DNA before and after each step of the replacement reaction as no net loss of nucleosomal species occurs during the histone replacement reaction *in vitro* (*Luk et al., 2010*).

While it is well established that H2A.Z is enriched at the promoter-proximal nucleosomes, it is underappreciated that substantial amount of H2A is also present at these sites (*Luk et al., 2010*). Experiments using sequential ChIP and tiling microarray analysis have demonstrated that in a population of G1-arrested cells, nucleosomes in the AA, AZ and ZZ configurations can all be detected at the +1 positions of most promoters (*Luk et al., 2010*). This observation suggests that the SWR1 reaction that generates ZZ nucleosomes is opposed by a pathway(s) that converts ZZ nucleosomes back to the AA state in a replication-independent manner. Consistent with this dynamic model, rapid, constitutive H3 turnover is observed at most +1 nucleosomes (*Dion et al., 2007*). Since the $(H3-H4)_2$ tetramer is at the center of the histone core (*Luger et al., 1997*), H3 turnover implies complete disassembly of the ZZ nucleosome (*Figure 1—figure supplement 1A*, step II). Reassembly likely leads to the formation of the canonical AA nucleosomes as H2A is ~10 times more abundant than H2A.Z and SWR1 does not assemble H2A.Z nucleosomes *de novo* on DNA (*Luk et al., 2010*; *West and Bonner, 1980*) (*Figure 1—figure supplement 1A*, step III).

The ATP-dependent remodeling complex INO80 has been reported to mediate the reverse replacement reaction (in which a nucleosomal H2A.Z-H2B dimer is replaced by a free H2A-H2B dimer) (*Figure 1—figure supplement 1A*, steps I-c and I-d) (*Papamichos-Chronakis et al., 2011*). Analysis of H2A.Z ChIP followed by microarray (ChIP-chip) showed that H2A.Z in *ino80Δ* cells redistributes from promoters to gene body regions as compared to wild-type cells (*Papamichos-Chronakis et al., 2011*). Deletion mutant of *ARP5*, a gene encoding a critical component of the INO80 complex, exhibited global H2A.Z accumulation especially around the promoters as demonstrated by the ChIP-exo technique (*Yen et al., 2013*). However, the ChIP-chip data of a more recent study disagreed, showing that the genome-wide H2A.Z occupancy was similar in *ino80Δ* and wild-type cells (*Jeronimo et al., 2015*). Therefore, what contributes to the conversion of ZZ nucleosomes to the AA state remains controversial.

This study addresses the hypothesis that the transcription machinery is a major driving force of the disassembly of the +1 H2A.Z nucleosomes. We used the anchor away approach to deplete components of the transcription machinery (*Haruki et al., 2008*) and a quantitative ChIP-seq approach to probe changes in H2A.Z occupancy genome-wide at single-basepair resolution. We observed reciprocal increase of H2A.Z and decrease of H2A genome-wide after depletion of the PIC. By contrast, nuclear depletion of Ino80 did not cause global H2A.Z accumulation. These findings suggest that the assembly of the Pol II transcription machinery and/or its activity contribute(s) to the constitutive turnover of H2A.Z at yeast promoters.

## Results

### A quantitative approach for measuring genome-wide levels of H2A.Z

The genomic H2A.Z level at any given promoter in a cell population is in a steady state that is maintained by the deposition mediated by SWR1 and the eviction mediated by putative chromatin remodeling pathway(s) (*Luk et al., 2010*). To test the contribution of the transcription machinery in H2A.Z eviction genome-wide, conditional yeast mutants were used to block the assembly of the PIC. If eviction of H2A.Z-containing nucleosomes is blocked, H2A.Z levels are expected to increase and H2A levels to decrease, as the SWR1 complex continues to replace nucleosomal H2A-H2B with

H2A.Z-H2B dimers (*Figure 1—figure supplement 1B*). To block PIC assembly, TBP was conditionally depleted from the nucleus using the anchor-away approach with *SPT15*, the gene that encodes TBP (*Haruki et al., 2008*). When fused to the FKBP12-rapamycin-binding domain (FRB), TBP-FRB can be dragged out of the nucleus in a rapamycin-dependent manner by the FKBP12 tag on the pre-ribosome (*Haruki et al., 2008*). Previous studies have shown yeast cells expressing *SPT15-FRB* (hereafter referred to as *TBP-FRB*) rapidly depleted TBP from Pol II promoters, blocked Pol II recruitment and shut off transcription within 1 hr of rapamycin treatment (*Grimaldi et al., 2014*; *Haruki et al., 2008*; *Wong et al., 2014*). In our experiments, TBP-FRB relocalized to the cytoplasm with similar kinetics (*Figure 1—figure supplement 2*).

To measure the relative occupancy of H2A.Z and H2A genome-wide, deep sequencing was combined with quantitative ChIP of H2A.Z (*Luk et al., 2010*). Specifically, chromatin from fixed haploid yeast cells expressing a 2xFLAG-epitope-tagged *HTZ1* gene was digested with micrococcal nuclease (MNase) to generate mononucleosomes. H2A.Z-containing nucleosomes were separated from the canonical AA nucleosomes by binding to anti-FLAG affinity gel followed by elution using FLAG peptides. Given that H2A.Z is the only H2A variant in budding yeast and the IP efficiency was consistently over 80% (*Figure 1—figure supplement 3*), the flow-through (FT) of the IP reaction was highly enriched for the homotypic AA nucleosomes (referred to as the H2A nucleosomes hereafter) and the IP fraction was enriched for the heterotypic AZ and homotypic ZZ nucleosomes (referred collectively to as the H2A.Z nucleosomes hereafter) (*Luk et al., 2010*). The DNA extracted from both the FT and IP fractions, as well as the DNA from the input nucleosomes used for the IP, was mapped by deep sequencing.

Forty-four reference regions (called no-Z-zones, covering 152,021 bp and 1,161 nucleosomes) with very low H2A.Z but high H2A occupancy were manually chosen and were used to normalize the FT fraction data to the input (*Figure 1—source data 1* and *Figure 1—figure supplement 4A*). Depletion of signal in the normalized FT fraction data relative to the input represents the immuno-depleted H2A.Z associated DNA. The amplitude of the H2A.Z data was adjusted using a curve fitting algorithm such that the sum of the normalized profiles of the H2A.Z fraction and FT fraction surrounding the +1 nucleosome region (N = 4,738) equals, to a first approximation, the input profile (*Luk et al., 2010*) (*Figure 1—figure supplement 4B*). This approach, called quantitative ChIP-seq or qChIP-seq, is similar to ChIP-coupled quantitative PCR (ChIP-qPCR) in that occupancy is reported in relation to the input of each ChIP reaction but is genome-wide. Unlike standard ChIP-seq, which typically involves normalization by equalizing the read counts of ChIP samples and reports ChIP signals in relation to some background or threshold (e.g. mean of genomic ChIP signal) that may vary among samples, the qChIP-seq method allows more quantitative comparison between samples, especially when a global change in H2A.Z occupancy is expected.

## PIC assembly is required for genome-wide H2A.Z eviction at yeast promoters

Using qChIP-seq, the coverage of nucleosomal DNA was determined for the H2A.Z (green), the FT (indicated as H2A in red), and the input (gray) fractions in the *TBP-FRB* haploid strain and the isogenic untagged control strain (no *FRB*). Under permissive conditions (no RAP), H2A.Z was most prevalent at the +1 nucleosomal positions (marked by '+'), less at the -1 positions and progressively less towards the 3' end of genes in agreement with published results (*Albert et al., 2007*) (*Figures 1A and B*, *top*, no RAP). This phenomenon is further demonstrated when the H2A.Z and input profiles were compiled at the dyads of 4,738 +1 nucleosomes (*Figures 1C and D*, *top left*). The FT fraction is a good representation of nucleosomal H2A levels, at least qualitatively, for two reasons. First, although the FT fraction may contain unstable histone-DNA complexes that are devoid of H2A, e.g. tetrasomes, or non-histone complexes (*Reja et al., 2015*), these structures are highly depleted in these experiments as the chromatin was subjected to extensive MNase digestion (unless indicated otherwise, below). Second, an earlier study that used an anti-H2A antibody to ChIP the FT fraction followed by high-resolution tiling microarray analysis produced an AA nucleosome profile that is highly similar to the FT profiles of our current experiments (*Luk et al., 2010*) (*Figures 1C and D*, *bottom left*). Therefore, the qChIP-seq data are consistent with previous conclusions that nucleosomes enriched for H2A dominate in coding regions and that, while depleted at the +1 positions, a substantial amount of H2A nucleosomes remains (*Figures 1C and D*, *bottom left*) (*Luk et al., 2010*). The FT profile is referred to as the H2A profile hereafter.

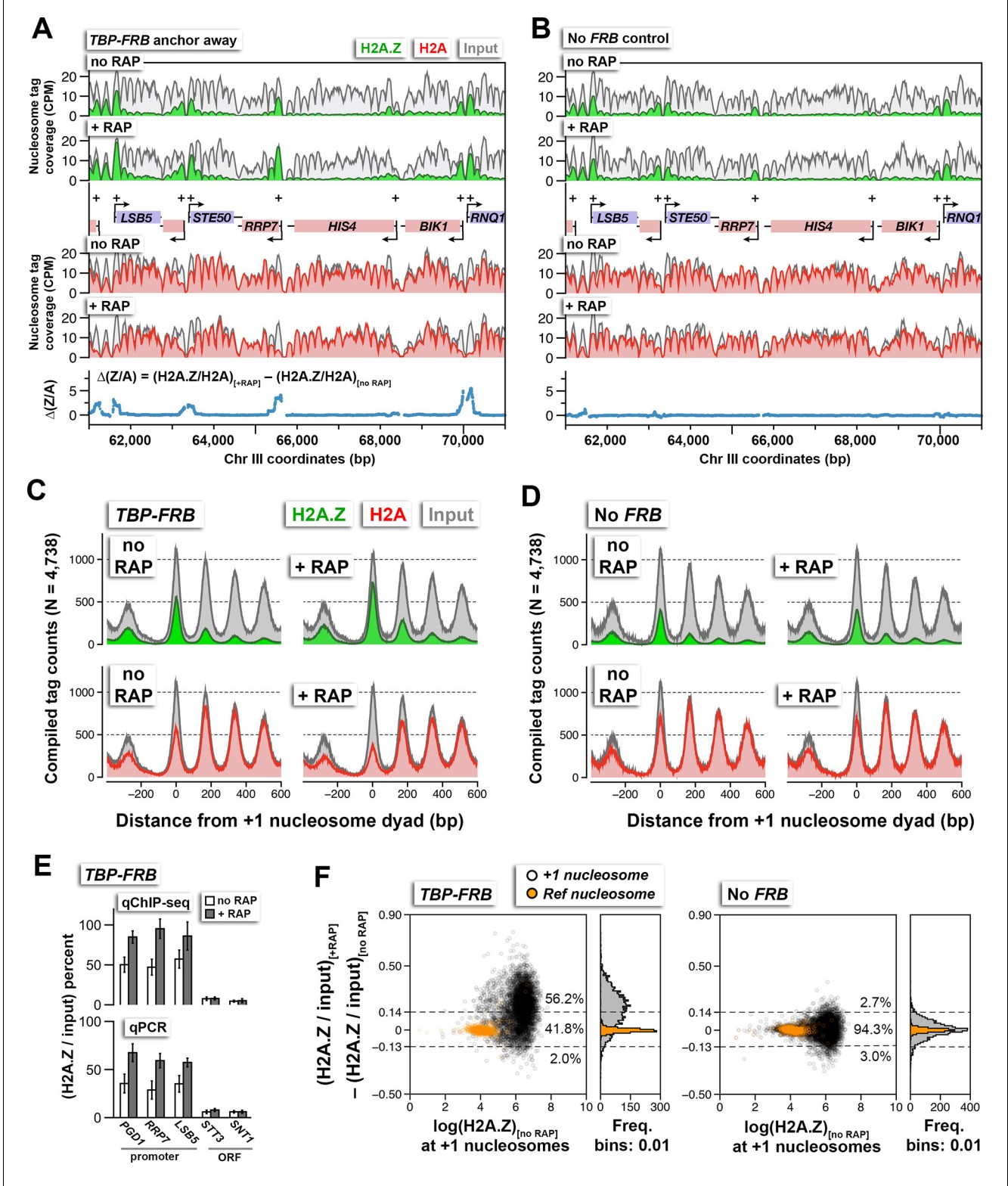

**Figure 1.** H2A.Z nucleosome occupancy determined by qChIP-seq in the *TBP-FRB* and the untagged control (no *FRB*) strains with and without rapamycin treatment. (A,B) Sequencing tag coverage of H2A.Z (in green), FT (indicated as H2A in red), and input (in gray) at a representative genomic region on chromosome III. *Blue traces* indicate Δ(Z/A), the H2A.Z-to-H2A (Z/A) ratio with rapamycin treatment (+RAP) minus that without treatment (no RAP). Plus signs and arrowheads mark +1 nucleosomes and transcription start sites, respectively. (C,D) Compiled read counts (midpoints) of H2A.Z (green), H2A (red), and input (gray) nucleosomes were centered around the dyad of 4,738 +1 nucleosomes. (E) Verification of the qChIP-seq

*Figure 1 continued on next page*

*Figure 1 continued*

data by qPCR using primer pairs covering the +1 nucleosomes of the indicated promoters and regions within the indicated open reading frames (ORF). (F) Scatter plots and histograms showing the change in (H2A.Z/input) of the +1 and reference nucleosomes as a function of endogenous H2A.Z level before rapamycin treatment. The (H2A.Z/input) value represents the ratio of H2A.Z tag coverage over input tag coverage within a 120 bp region around the nucleosome dyad. *Open black circles* mark the +1 nucleosomes (*Rhee et al., 2014*). *Orange dots* mark the reference nucleosomes used for normalization. *Dotted lines* represent the upper and lower thresholds for significant change in H2A.Z levels, which are defined as two standard deviations from the median of the no *FRB* control. The percentages of data points within and outside the threshold regions are indicated. The qChIP-seq and qPCR data represent averages of >3 independent ChIP reactions (technical replicates) of two independent cultures (biological replicates). The error bars in (E) represent standard deviation.

The following source data and figure supplements are available for figure 1:

**Source data 1.** Reference regions or 'no-Z-zones' used for normalization.

**Source data 2.** Normalization factors.

**Source data 3.** Nucleosome tag profiles of *TBP-FRB* around the +1 dyads before normalization.

**Source data 4.** Nucleosome tag profiles of *no FRB* around the +1 dyads before normalization.

**Source data 5.** Nucleosome tag profiles of *RPB1-FRB* around the +1 dyads before normalization.

**Source data 6.** Average tag coverage around the +1 dyads of *TBP-FRB*.

**Source data 7.** Average tag coverage around the +1 dyads of no *FRB*.

**Source data 8.** Average tag coverage around the +1 dyads of *RPB1-FRB*.

**Source data 9.** Source data for *Figure 1E*.

**Figure supplement 1.** A proposed model to account for the constitutive histone turnover at yeast promoters.

**Figure supplement 2.** Fluorescence microscopy of yeast expressing *TBP-FRB-GFP* with and without rapamycin treatment.

**Figure supplement 3.** Immunoblot analysis to control for anti-FLAG IP efficiency.

**Figure supplement 4.** The strategy used to normalize relative H2A and H2A.Z occupancy.

**Figure supplement 5.** Heatmaps of H2A.Z, H2A and input nucleosomes for the *TBP-FRB*, *no FRB*, and *RPB1-FRB* strains .

**Figure supplement 6.** Concordance of relative H2A.Z occupancy between biological replicates.

**Figure supplement 7.** Using Δ(Z/A) as a parameter to identify +1 nucleosomes with PIC-dependent H2A.Z eviction.

**Figure supplement 8.** Relative H2A.Z occupancy before and after Rpb1 depletion.

**Figure supplement 9.** H2A.Z accumulation is not due to aberrant accumulation of the SWR1 complex.

**Figure supplement 10.** Input nucleosome occupancy at the +1 positions before and after TBP depletion.

**Figure supplement 11.** qPCR and H2A western analyses of the flow-through fractions.

Rapamycin treatment of *TBP-FRB* cells for 1 hr resulted in H2A.Z accumulation at the promoter-proximal nucleosomes of most genes, with a corresponding depletion of H2A signal (compare + RAP and no RAP in *Figures 1A and C* and in *Figure 1—figure supplement 5A*). Rapamycin treatment alone did not cause significant change in histone dynamics as the untagged wild-type cells exhibited similar promoter-specific H2A.Z levels before and after rapamycin treatment

(*Figures 1B and D*, *Figure 1—figure supplement 5B*). To confirm that the increase of H2A.Z levels in the *TBP-FRB* strain was not an artifact of normalization, qPCR was employed to measure the immunoprecipitated DNA in the H2A.Z fraction relative to the input at three +1 nucleosome regions and two coding regions. Similar to the sequencing analysis, the qPCR experiments indicated an increase in H2A.Z nucleosomal DNA at +1 nucleosomes after depletion of TBP, with no change observed for DNA located within the open reading frames (ORFs), confirming the robustness of the qChIP-seq approach (*Figure 1E*).

To evaluate the H2A.Z change at the +1 positions genome-wide, the change in (H2A.Z/input) ratios at 4,738 +1 nucleosomes after TBP depletion {i.e. $(H2A.Z/input)_{[RAP]} - (H2A.Z/input)_{[no\ RAP]}$} were plotted against the endogenous H2A.Z levels represented by the logarithmically transformed H2A.Z tag counts before rapamycin treatment (*Figure 1F*). A threshold for significant change in (H2A.Z/input) was defined by two standard deviations above and below the median of the untagged control (*Figure 1F*, dotted lines). The change in (H2A.Z/input) values of the reference nucleosomes used in normalization was plotted for comparison (*Figure 1F*, *orange dots*). Fifty-six percent of +1 nucleosomes exhibited a significant increase in relative H2A.Z signal upon TBP depletion (*Figure 1F*). The (H2A.Z/input) signals of the two biological replicates were highly reproducible (*Figure 1—figure supplement 6A–B*). Note the global shift of data points towards the upper right quadrant in the +RAP data as compared to the no RAP sample, indicating a global increase of relative H2A.Z in both biological replicates (compare *Figure 1—figure supplement 6A–B*). By contrast, similar shift of data points towards the upper right quadrant was not observed for the no FRB control (*Figure 1—figure supplement 6C–D*).

It is noteworthy that under the permissive condition, the *TBP-FRB* strain exhibited a higher endogenous H2A.Z occupancy compared to the untagged strain (compare no RAP in *Figures 1C and D*). The difference likely reflects a partial defect of the *TBP-FRB* allele that has predisposed the cells to H2A.Z accumulation. But importantly, the measurement of *change* in the (H2A.Z/input) ratio (*Figure 1F*) normalizes any differences in the ground state (no RAP) and highlights the functional consequence of the depletion of TBP or other factors in question (below).

The change in (H2A.Z/H2A) ratios {i.e. $(H2A.Z/H2A)_{[RAP]} - (H2A.Z/H2A)_{[no\ RAP]}$}, which is referred to as Δ(Z/A) hereafter, provides an even more sensitive, nonetheless non-linear, indicator of H2A.Z dynamics because of the antagonistic change in H2A.Z and H2A occupancy. As shown in *Figure 1—figure supplement 7A*, 72% +1 nucleosomes exhibited an increase in Δ(Z/A) signal upon TBP depletion. When the Δ(Z/A) values were plotted along the chromosome coordinates, this parameter clearly identified sites with strong H2A.Z dynamics (*Figure 1A*, *blue traces*). The Δ(Z/A) parameter will later be used to identify novel transcription start sites.

To test if depletion of another component of the PIC could also lead to H2A.Z accumulation, we targeted Rpb1, the largest subunit of Pol II, for nuclear depletion by anchor-away. The *RBP1-FRB* construct has previously been shown to effectively shut off transcription (*Haruki et al., 2008*). Similar to the results of TBP-FRB removal, nuclear depletion of Rpb1-FRB led to strong H2A.Z accumulation and H2A depletion at most promoters (*Figure 1—figure supplement 8* and *Figure 1—figure supplement 5C*).

## H2A.Z accumulation in response to PIC depletion is not due to aberrant accumulation of the SWR1 complex

One explanation for the increase in H2A.Z levels at the +1 promoter upon TBP depletion is that the balance between H2A.Z eviction and deposition of H2A.Z nucleosomes by SWR1 has been disrupted. SWR1 continues to convert H2A nucleosomes to the H2A.Z containing forms but with no eviction to restore these nucleosomes back to the AA state in the absence of the PIC. Consistent with this idea, sites with strong H2A.Z dynamics are more enriched for endogenous SWR1 (*Figure 1—figure supplement 9A*). An alternative explanation for this phenomenon, however, is that TBP depletion does not block H2A.Z eviction but instead leads to recruitment of aberrantly high levels of SWR1 to promoters as the PICs dissociate. This model predicts that SWR1 levels should increase at promoters upon TBP depletion. To distinguish between these two possibilities, ChIP-qPCR was used to monitor the occupancy of the Swr1 subunit in the *TBP-FRB* strain at different times after rapamycin treatment. The promoter regions of *SWR1* itself and *FUN12* were chosen because these sites are known to be enriched for SWR1 (*Ranjan et al., 2013*; *Yoshida et al., 2010*) and exhibited strong changes in H2A.Z dynamics upon TBP depletion (*Figure 1—figure supplement*

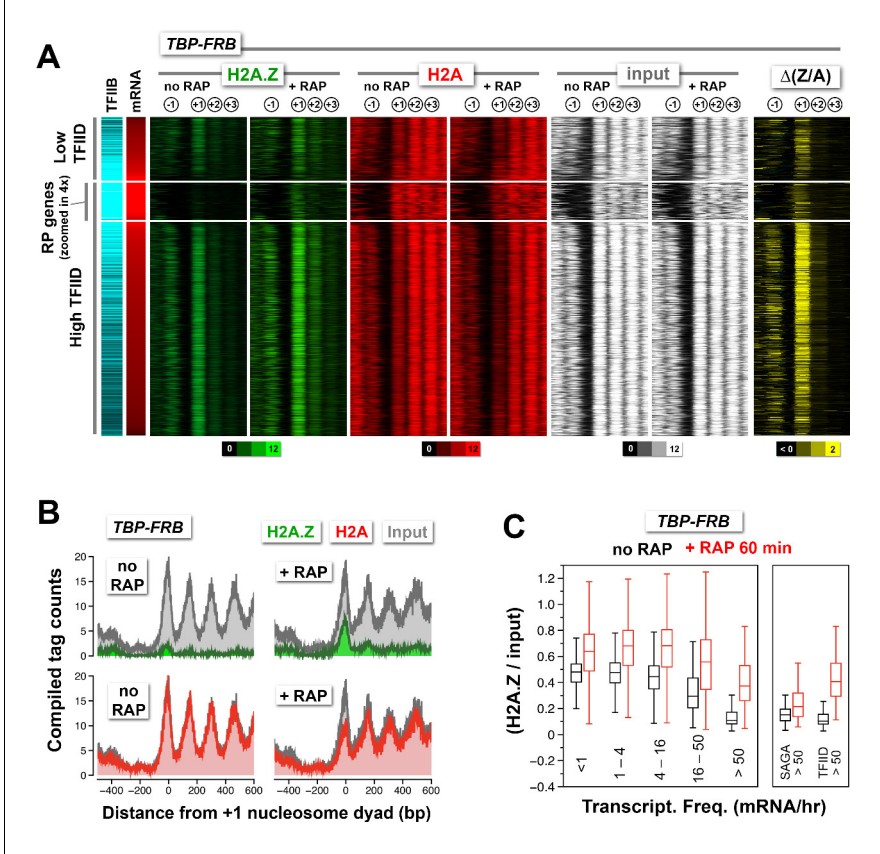

**Figure 2.** Change in nucleosomal H2A.Z and H2A levels near promoters before and after depletion of TBP. (**A**) Heatmaps showing the average normalized tag coverage of H2A.Z (*green*), H2A (*red*) and input (*white*), and the corresponding Δ(Z/A) values (*yellow*) around the +1 dyads of the *TBP-FRB* strain with and without rapamycin treatment. Promoters enriched for the PIC (based on Sua7 occupancy) were grouped into high and low TFIID (based on Taf1 occupancy) and were then sorted by mRNA abundance (n = 3,919) (*Lipson et al., 2009*; *Rhee and Pugh, 2012*). The ribosomal protein (RP) genes (n = 128) are 'zoomed in 4x' meaning that the line thickness is 4 times of the other genes. (**B**) Compiled nucleosome tag counts of H2A.Z (green), H2A (red), and input (gray) around the +1 dyads of the RP genes. (**C**) The +1 nucleosomes were grouped according to transcriptional frequency (*Holstege et al., 1998*) and the (H2A.Z/input) values were presented as box plots. *Box*: interquartile range (IQR); *line in box:* median; *whiskers*: range. The most active genes (>50 mRNA/hr) were sub-divided into two groups that are SAGA- or TFIID-enriched. The number of +1 nucleosomes in the transcriptional frequency groups <1, 1–4, 4–16, 16–50, and >50 mRNA/hr are 937, 1932, 1018, 219 and 161, respectively. There are 22 promoters in the SAGA >50 group and 130 in the TFIID >50 group.

The following figure supplement is available for figure 2:

**Figure supplement 1.** Normalized nucleosome tag coverage for H2A.Z, H2A, and input before and after the depletion of Rpb1.

*9B*). Rather than accumulating at the promoters of these two genes, SWR1 dissociated shortly after TBP-FRB depletion as demonstrated by the decrease in Swr1 ChIP signal (*Figure 1—figure supplement 9C*). This observation supports the idea that the PIC is required to actively evict H2A.Z nucleosomes. In addition, the coincidental accumulation of H2A.Z and depletion of SWR1 suggest that endogenous SWR1 dissociates from promoters after depositing H2A.Z. This is in agreement with the *in vitro* observation that SWR1 has a lower affinity for the homotypic H2A.Z nucleosome product than the homotypic H2A nucleosome substrate (*Ranjan et al., 2015*).

## Constitutive PIC-dependent H2A.Z eviction is associated with promoters of active and infrequently transcribed genes that are generally TFIID enriched

Yeast promoters that are TFIID-enriched or -depleted are apparently regulated by different mechanisms (*Rhee and Pugh, 2012*). Although both types of promoters require TBP for transcription,

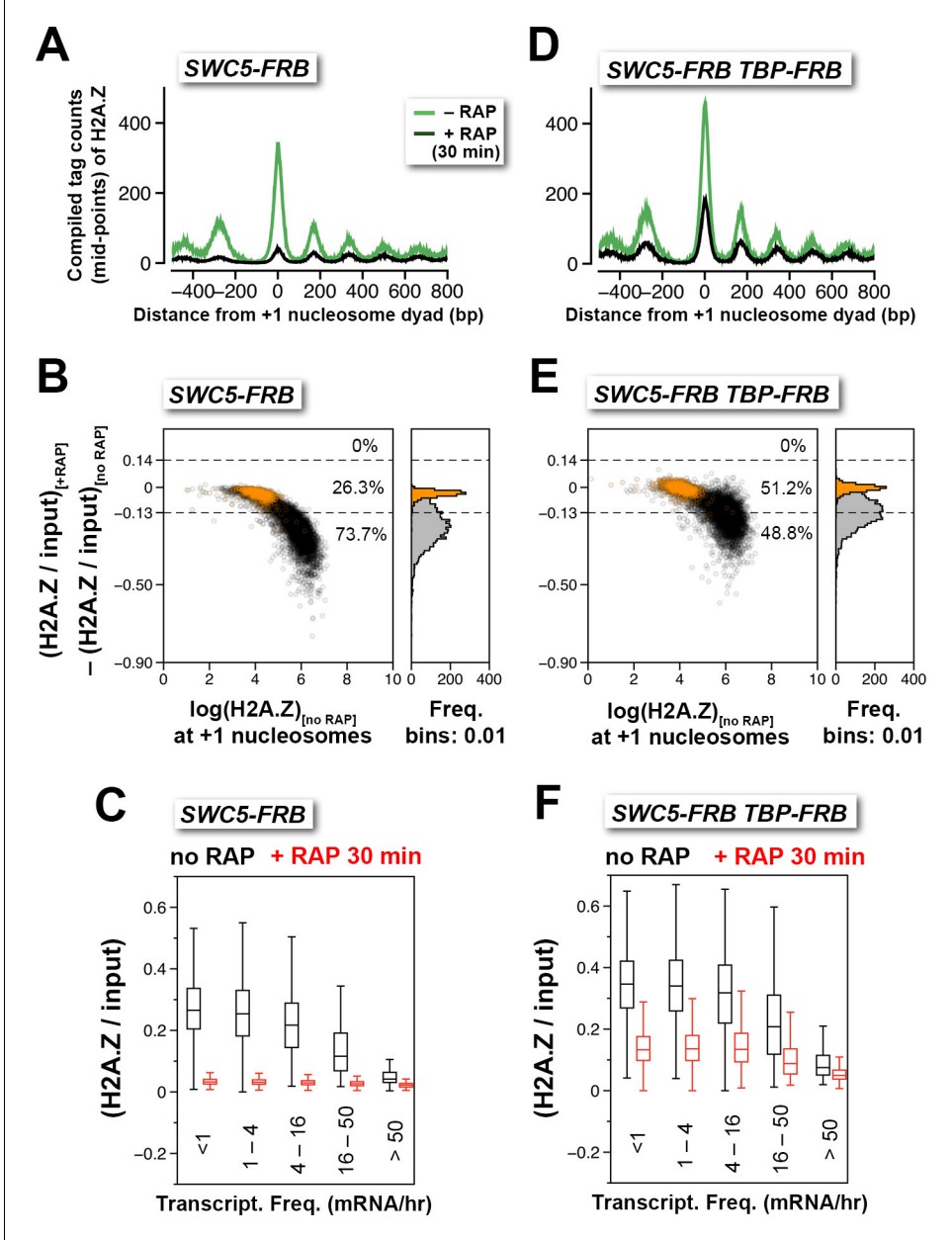

**Figure 3.** Depletion of Swc5 revealed rapid PIC-dependent eviction of H2A.Z at the +1 nucleosome of active and infrequently transcribed genes. (**A**) Compiled tag counts of H2A.Z nucleosomes around the +1 dyads in the *SWC5-FRB* strain before and after 30 min of rapamycin treatment. (**B**) Scatter plot analysis showing the change in relative H2A.Z occupancy against endogenous H2A.Z level in the *SWC5-FRB* strain at the +1 nucleosomes (n = 4,738) after 30 min of rapamycin treatment. *Gray*: +1 nucleosomes. *Orange*: reference nucleosomes depleted for H2A.Z. (**C**) Same as *Figure 2C*, except the *SWC5-FRB* strain was used. (**D–F**) Same as (**A–C**), except the *SWC5-FRB TBP-FRB* strain was used.

The following source data and figure supplements are available for figure 3:

**Source data 1.** Compiled nucleosome tag profiles of *SWC5-FRB* and *SWC5-FRB TBP-FRB* around the +1 dyads before normalization.

**Source data 2.** Average tag coverage around the +1 dyads of *SWC5-FRB*.

**Source data 3.** Average tag coverage around the +1 dyads of *SWC5-FRB TBP-FRB*.

**Figure supplement 1.** Concordance of relative H2A.Z occupancy between technical replicates (independent IP reactions).

*Figure 3 continued*

**Figure supplement 2.** Relative H2A.Z occupancy at the +1 nucleosomes of the *SWC5-FRB* strain at different times after rapamycin treatment.

TFIID-enriched genes are often TATA-less and associated with a more open NDR, whereas TFIID-depleted promoters generally contain a consensus TATA element and are more covered with nucleosomes and more enriched with the SAGA complex (*Basehoar et al., 2004*; *Rhee and Pugh, 2012*). To see if the +1 nucleosomes at these two types of promoters exhibit differential H2A.Z dynamics, the occupancy of H2A, H2A.Z and input nucleosomes, and the corresponding Δ(Z/A) values, before and after rapamycin treatment were sorted by TFIID enrichment and transcript abundance (*Lipson et al., 2009*; *Rhee and Pugh, 2012*). As seen in *Figure 2A*, the +1 nucleosomes that exhibited stronger H2A.Z accumulation after TBP depletion are generally more enriched for TFIID.

The ribosomal protein (RP) genes are regulated by TFIID and are among the most highly transcribed (*Rhee and Pugh, 2012*; *Warner, 1999*). Their promoters are unusual in that the endogenous +1 nucleosomes are generally depleted for H2A.Z but relatively enriched for H2A (*Figure 2B*, n = 128). Upon depletion of TBP, relative H2A.Z accumulates dramatically at the +1 position (*Figure 2B*, green). The data suggest that H2A.Z deposition occurs at the RP gene promoters but that H2A.Z nucleosomes are quickly removed by a mechanism that is dependent on TBP, leading to a low steady-state H2A.Z occupancy. Interestingly, Rpb1 depletion led to almost no change in H2A.Z occupancy at the +1 nucleosomes of RP genes (*Figure 2—figure supplement 1*). Since RP genes promoters are occupied by high level of PIC components (indicated by the TFIIB marker Sua7), partial PIC may remain bound after Rpb1 depletion. This suggests that H2A.Z removal at the RP genes requires TBP but not Pol II.

To further understand the link between transcriptional activity and H2A.Z dynamics, the +1 H2A.Z occupancy of 4,267 promoters before and after TBP depletion was sorted and grouped by the transcriptional frequency of their downstream genes and compared by box plot analysis (*Figure 2C*) (*Holstege et al., 1998*). Before TBP-depletion, the +1 nucleosomes associated with moderately or infrequently transcribing genes exhibited >40% (median) relative H2A.Z occupancy for genes with <16 mRNA/hr (n = 3,887) and ~30% (median) for those with 16–50 mRNA/hr (n = 219). For the top 3% of most highly transcribing genes (>50 mRNA/hr, n = 161) the steady-state H2A.Z occupancy is ~11% (median) before TBP-depletion. Upon rapamycin treatment, substantial increase of relative H2A.Z occupancy is observed in all groups (*Figure 2C*). The increase is more dramatic in the most highly transcribing genes, suggesting that the low steady-state H2A.Z occupancy is due to strong transcriptional activity (*Figure 2C*, >50 mRNA/hr). Interestingly, when these highly transcribing genes were sorted based on the enrichment of SAGA or TFIID (*Basehoar et al., 2004*), H2A.Z accumulated to a smaller extent for the SAGA-enriched genes than for the TFIID-enriched genes (*Figure 2C*, *right*). Therefore, unlike the strong TFIID-enriched promoters where low steady-state +1 H2A.Z level is due to robust PIC-dependent H2A.Z eviction activity, low H2A.Z at the strong SAGA-enriched promoters is due to either strong PIC-independent eviction or weak SWR1-mediated H2A.Z deposition or both.

An alternative approach to monitor H2A.Z eviction is to conditionally block H2A.Z deposition and follow the depletion of H2A.Z. Since SWR1-mediated H2A.Z deposition requires Swc5, a subunit of the SWR1 complex (*Wu et al., 2005*), anchor-away was utilized to deplete Swc5 so as to conditionally block H2A.Z deposition. SWR1 activity of the *SWC5-FRB* strain was strongly inhibited after 30 min of rapamycin treatment as demonstrated by the robust loss of H2A.Z occupancy (*Figures 3A–B* and *Figure 3—figure supplement 1A–B*). When the relative H2A.Z levels (H2A.Z/input) of the +1 nucleosomes were sorted and grouped by the transcriptional frequency of the downstream genes, almost background level of H2A.Z was observed in all groups, indicating that robust, constitutive H2A.Z eviction at the +1 position occurs at both active and infrequently transcribed genes (*Figure 3C*). To further understand how fast is the eviction of H2A.Z, relative H2A.Z levels were measured at various time points after Swc5 depletion (*Figure 3—figure supplement 2*). At most +1 positions, relative H2A.Z levels dropped below 50% of endogenous levels after 15 min of rapamycin treatment, indicating that the occupancy half-life of H2A.Z is less than 15 min. Since virtually baseline level of H2A.Z remains after 60 min of rapamycin treatment (*Figure 3—figure supplement 2*, *red*),

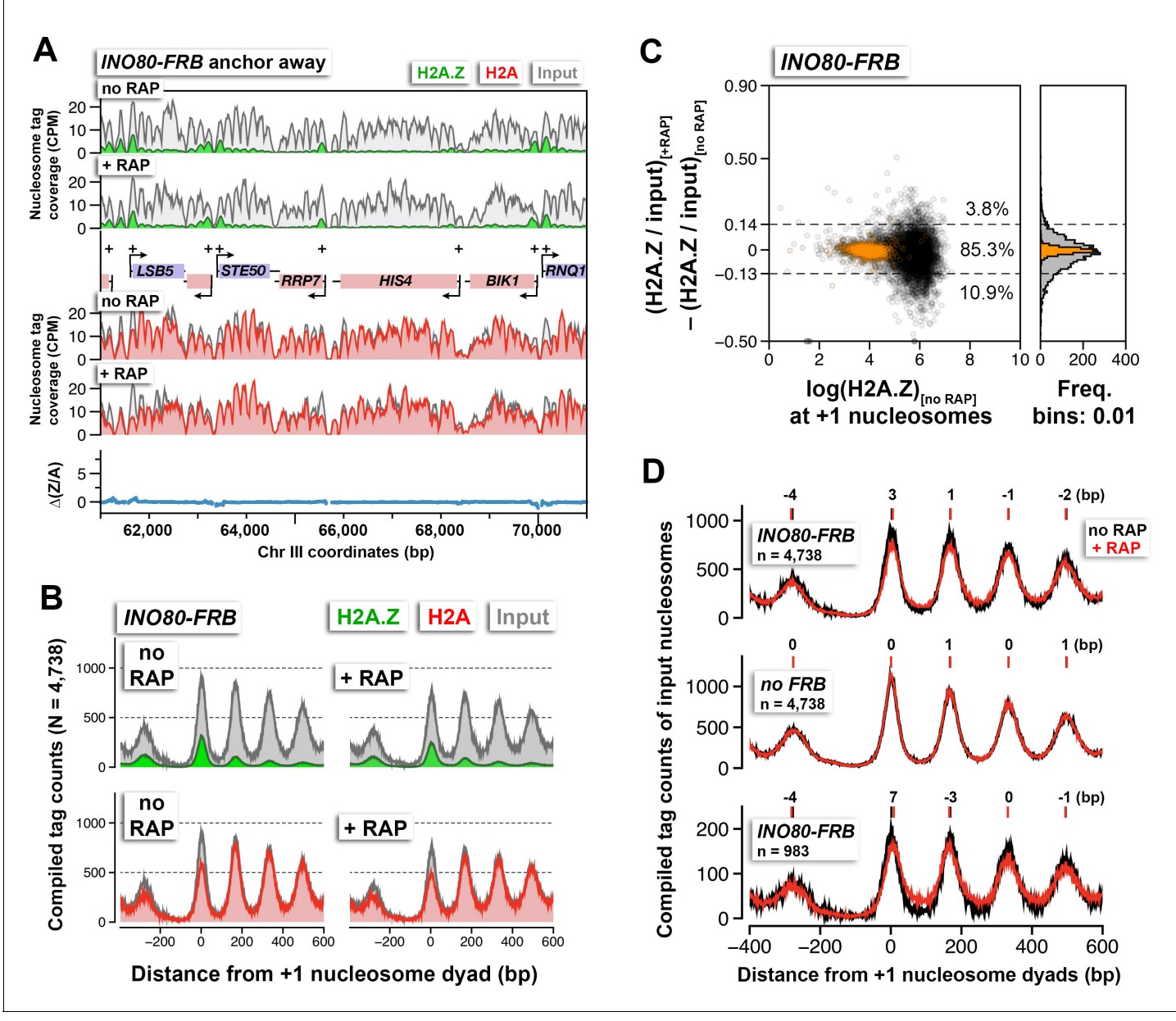

**Figure 4.** The effects of Ino80 depletion on H2A.Z occupancy and nucleosomal positions. (A) Sequencing tag coverage of H2A.Z (*green*), H2A (*red*), and input (*gray*) nucleosomes, as well as the corresponding Δ(Z/A) values were plotted as described in *Figure 1A*. (B) Compiled read counts around the +1 nucleosome dyads. (C) Scatter plot and histogram showing the change in (H2A.Z/input) for individual +1 nucleosomes after Ino80 depletion. The thresholds (*dotted lines*) were determined as described in *Figure 1F*. Open black circles: +1 nucleosomes. *Orange*: reference nucleosomes. (D) The compiled input nucleosome profiles before (*black*) and after (*red*) rapamycin treatment in *INO80-FRB (top*) and no *FRB, (middle*) were re-plotted using the data from (B) and *Figure 1D* to highlight any difference in the nucleosomal arrays. *Bottom*: The compiled tag counts of 983 genes with fuzzier nucleosomal organization upon Ino80 depletion. The integers above the nucleosomal peaks indicate shift distance in base pairs. Positive values indicate right-shift after rapamycin treatment and *vice versa*. Peak center positions were determined by curve fitting with a Gaussian model followed by local maxima calculation. All qChIP-seq data of *Ino80-FRB* represent the mean of two biological replicates.

The following source data and figure supplement are available for figure 4:

**Source data 1.** Nucleosome tag profiles of *INO80-FRB* around the +1 dyads before normalization.
**Source data 2.** Average tag coverage around the +1 dyads of *INO80-FRB*.
**Source data 3.** The list of 983 genes with fuzzier nucleosomal organization after Ino80 depletion.

*Figure 4 continued on next page*

*Figure 4 continued*

**Figure supplement 1.** Verification of the conditional depletion of Ino80-FRB and concordance of biological replicates.

the data also suggests that SWR1 is the sole deposition pathway of H2A.Z at budding yeast promoters.

Next, we tested whether TBP-FRB depletion can slow H2A.Z eviction while the H2A.Z deposition activity of SWR1 is inhibited. Consistent with the idea that the PIC is required for H2A.Z eviction, depleting both TBP and Swc5 in the double mutant resulted in a less dramatic decrease in relative H2A.Z occupancy at most +1 nucleosomes as compared to the single *SWC5-FRB* mutant for the same duration (i.e. 30 min) of rapamycin treatment (*Figures 3D–F* and *Figure 3—figure supplement 1C–D*). It is noteworthy that TBP depletion did not completely prevent the depletion of H2A.Z caused by Swc5 depletion. One explanation is that the kinetics of TBP depletion by anchor away is slower than that of Swc5. Alternatively, a PIC-independent H2A.Z eviction pathway might be operating.

## INO80 cannot account for bulk H2A.Z dynamics

The INO80 complex has been reported to catalyze the reverse H2A.Z replacement reaction in which nucleosomal H2A.Z-H2B dimers are replaced with free H2A-H2B dimers (*Papamichos-Chronakis et al., 2011*). Therefore, inactivation of the *INO80* gene (which encodes the catalytic core subunit of the INO80 complex) in cells with intact SWR1 activity is expected to accumulate H2A.Z. Endogenous *INO80* was fused to a tandem *FRB-GFP* tag to allow conditional depletion by anchor-away and visualization by fluorescence microscopy. Mutants defective for *INO80* function are hyper-sensitive to hydroxyurea (*Shen et al., 2003*). Cells with *INO80-FRB-GFP* (referred to as *INO80-FRB* hereafter) exhibited slow growth in medium containing both hydroxyurea and rapamycin but not hydroxyurea alone, confirming that *INO80* function can be abolished in a rapamycin-dependent manner (*Figure 4—figure supplement 1A*). As expected, in the absence of rapamycin, Ino80 was found in the nucleus (*Huh et al., 2003*) (*Figure 4—figure supplement 1B–C* at 0 min +RAP). After 90 min of rapamycin treatment, Ino80 was largely dispersed from the nuclei (*Figure 4—figure supplement 1C*). These cells were then fixed by formaldehyde crosslinking and the H2A.Z and H2A levels were measured by qChIP-seq. No significant global increase of H2A.Z was observed after Ino80 depletion (*Figures 4A–C* and *Figure 4—figure supplement 1D*). Instead, nucleosomal arrays became 'fuzzier' as demonstrated by the general decrease of nucleosomal peak height and the decrease of valley depth at the linker regions (*Figure 4D*, *top*). By contrast, in the untagged control (no *FRB*), the density and positions of the nucleosomal arrays before and after rapamycin treatment were unchanged (*Figure 4D*, *middle*). Using the difference in nucleosomal density within the linker region between +1 and +2 as a criterion for fuzziness, the chromatin arrays of 983 genes that require Ino80 for positioning were identified by clustering analysis (k = 3). The compiled chromatin arrays of these promoters showed, upon Ino80 depletion, the -1 and +1 nucleosomes shifted away from the NDR (*Figure 4D*, *bottom*). Importantly, the direction of shift at these nucleosomal positions are consistent with the published results of *ino80Δ* and are reproducible in both biological replicates (*Yen et al., 2012*) (*Figure 4—figure supplement 1E*). Overall, our data is in agreement with the INO80 remodeler functioning as a histone octamer slider but not an evictor of H2A.Z nucleosomes *in vivo* (*Jeronimo et al., 2015*; *Shen et al., 2003*; *Yao et al., 2016*; *Yen et al., 2012*).

## NDR formation is upstream of PIC assembly

The nuclear depletion experiments of TBP and Rpb1 also provide insights into the role of the transcription machinery in nucleosomal spacing organization. Both TBP and Rpb1 depletion caused histone octamers to reposition away from the NDR (*Figure 5A*). By contrast, no significant positional shift was observed in the nucleosomal arrays of the control cells (*Figure 4D*, no *FRB*). Similar results were previously observed by inactivating the transcription machinery with the *rpb1-1* mutant, which contains a temperature sensitive allele of *RPB1* (*Weiner et al., 2010*). These data are in agreement with the *in vitro* data that showed an elongating Pol II can disassemble the nucleosome in its path,

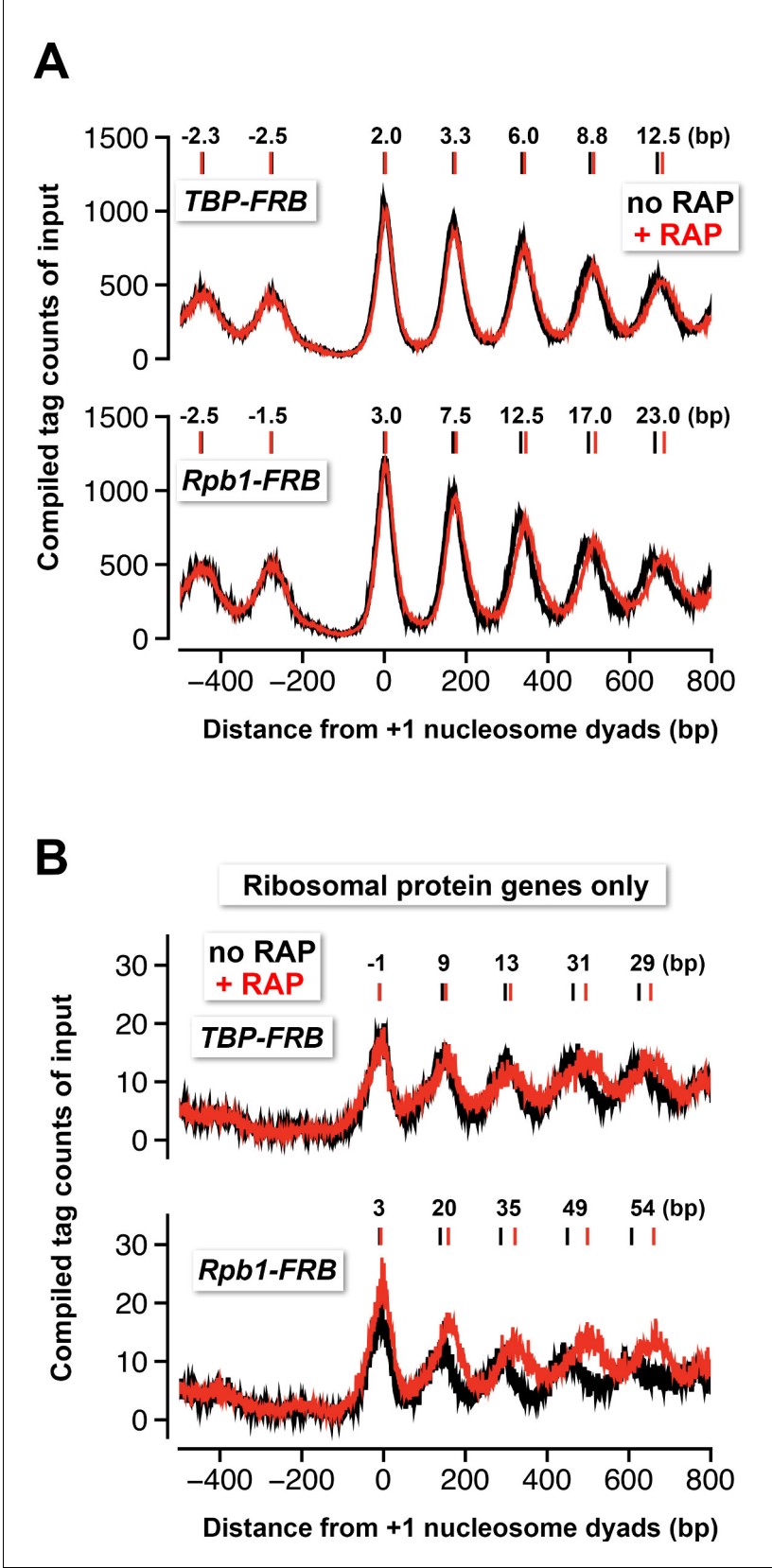

**Figure 5.** Nucleosomes shift away from the NDR in response to TBP and Rpb1 depletion. (**A**) The compiled tag counts of the input nucleosomal fraction (n = 4,738) before (*black*) and after (*red*) nuclear depletion of TBP-FRB

*Figure 5 continued on next page*

*Figure 5 continued*

(*top*) and Rpb1-FRB (*bottom*) were re-plotted using the data from *Figure 1* and *Figure 1C—figure supplement 8B*. The integers above the nucleosomal peaks indicate shift distance after rapamycin treatment. Peak center positions were determined as described in *Figure 4D*. (B) Same as (A) except that the compiled data of the ribosomal protein genes (n = 128) are shown.

while reassembling the nucleosome at a position slightly more upstream (*Clark and Felsenfeld, 1992*). But unlike the *rpb1-1* data, which exhibited strong downstream shift in the +1 nucleosomes under the non-permissive condition, the +1 shift upon TBP and Rpb1 depletion was comparatively minor (*Weiner et al., 2010*) (*Figure 5A*). In fact, the nucleosomal positional shift is progressively more dramatic towards the 3' end of the genes in the TBP and Rpb1 depletion experiments (*Figure 5A*). Interestingly, the positional shift caused by Rpb1 depletion is greater than that caused by TBP depletion (*Figure 5A*). One explanation for the difference is that TBP depletion blocks the Pol II molecules that are initiating but not those that have already engaged in elongation, whereas Rpb1 depletion removes all Pol II from the genome. But in both cases, the size of the NDR remains largely unchanged before and after the depletion of TBP or Rpb1 indicating that the formation of the NDR can be established in the absence of the PIC.

The phenomenon of nucleosome downshift in gene body is exaggerated at the RP genes where Pol II-mediated transcriptional activity is very high (*Figure 5B*) (*Warner, 1999*). But again, the positional shift of the +1 nucleosomes is comparatively insignificant (*Figure 5B*). This observation is important as a recent study showed repression of the RP genes by heat shock can cause their +1 nucleosomes to shift tens of basepairs in the upstream direction, indicating that these +1 nucleosomes are normally pushed downstream when the RP genes are actively transcribed (*Reja et al., 2015*). Our data showed that PIC is not responsible for the downstream push of the +1 nucleosomes and suggests that a chromatin remodeling enzyme(s) might be responsible for setting up the NDR before PIC assembly.

## Sites of strong H2A.Z dynamics are restricted to the +1 nucleosome of Pol II-transcribed genes, not -1 or fragile nucleosomes

Robust H2A.Z dynamics is generally associated with the +1 nucleosomal position and, to lesser extent, the -1 position [see Δ(Z/A) in *Figure 2A*, *right*]. However, since many yeast genes are oriented divergently, the H2A.Z dynamics observed at the -1 nucleosome of some promoters could be the +1 nucleosome of a divergent promoter. Indeed, when promoters were sorted based on the orientation of the upstream gene, positive Δ(Z/A) values were more often seen in the upstream region of a promoter with a divergent-oriented gene (*Head-Head*) than with a tandem-oriented gene (*Head-Tail*) (*Venters and Pugh, 2009*) (*Figure 6A*). The difference in H2A.Z dynamics at the +1 and -1 positions is exemplified by 44 divergent promoters that are separated by a *bona fide* -1 nucleosome (*Figure 6—figure supplement 1*). At these sites, H2A.Z accumulation was restricted to the +1 positions but not the -1 positions.

In addition to Pol II-transcribed genes, TBP is also required for transcription by Pol I and Pol III (*Cormack and Struhl, 1992*; *Schultz et al., 1992*). An earlier study showed that TBP-FRB can be depleted from Pol I and Pol III promoters to ~60% and ~10% of the original levels, respectively, using the same duration of rapamycin treatment (*Grimaldi et al., 2014*). Contrastingly, at the Pol I-dependent *RDN37* promoter, depletion of TBP-FRB actually caused modest H2A.Z depletion and H2A accumulation (*Figure 6B*). The Pol III-dependent tRNA genes have intragenic promoters (*Schramm and Hernandez, 2002*). These genes are typically flanked by nucleosomes with low H2A.Z content (*Albert et al., 2007*). After TBP depletion, the flanking nucleosomes showed only a subtle increase in H2A.Z (*Figure 6C*). However, there is a substantial encroachment of nucleosome density (presumably H2A-containing) within the tRNA gene-associated NDRs (*Figures 6C–D*). These results suggest that the assembly or activity of the Pol III transcription machinery is important for creating the NDRs at these sites.

The NDR region is not completely histone free but is occasionally associated with unstable, histone-containing structures, also known as 'fragile nucleosomes', that are hypersensitive to MNase digestion (*Kubik et al., 2015*; *Weiner et al., 2010*; *Xi et al., 2011*). An outstanding question is

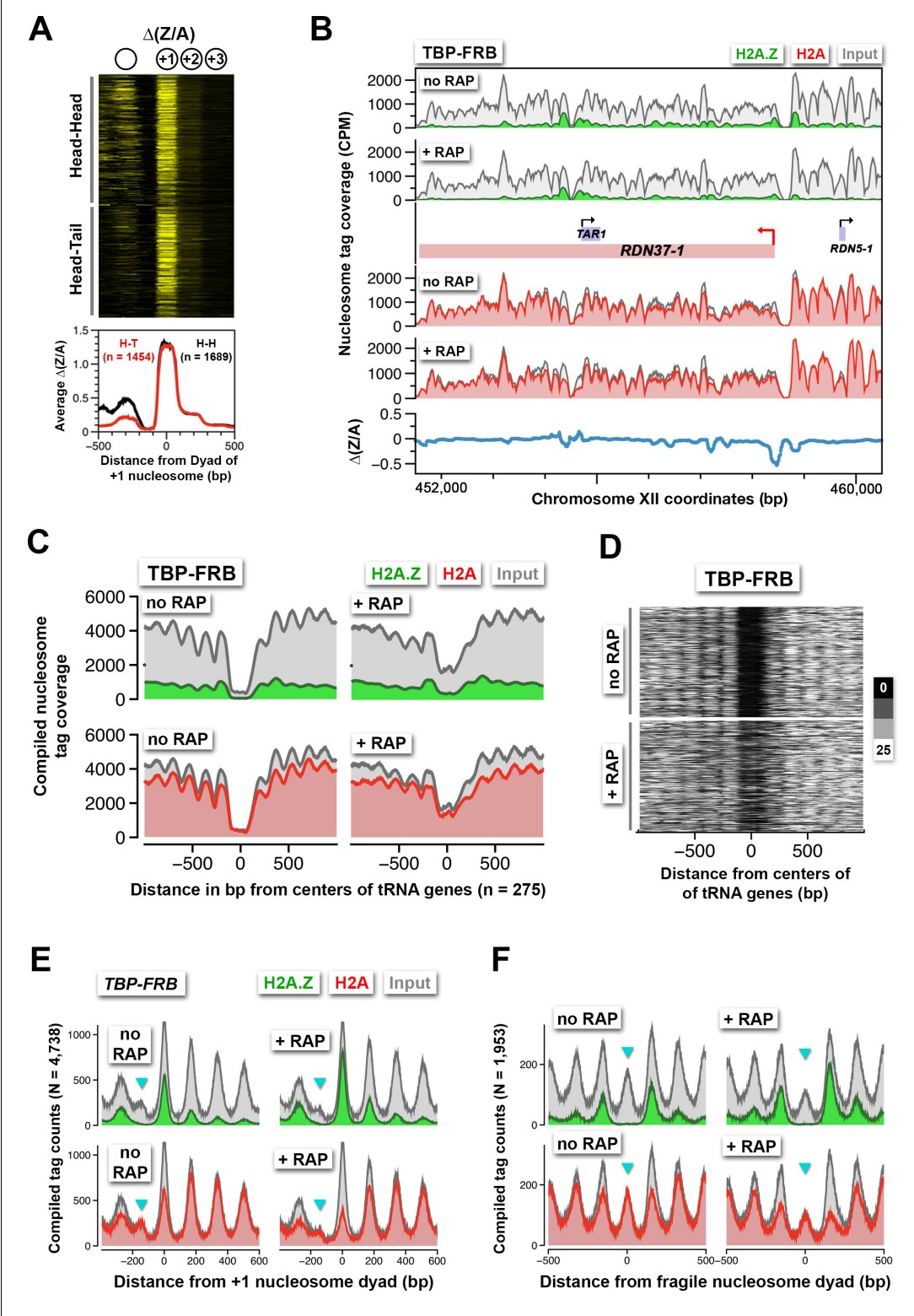

**Figure 6.** Rapid H2A.Z dynamics is restricted to the +1 nucleosomes of Pol II promoters, not to -1 nucleosomes or fragile nucleosomes. (A) A heatmap showing the Δ(Z/A) values of *TBP-FRB* around the +1 dyads of 3,143 promoters sorted by the orientation of the upstream gene. *Head-Head (H-H)*: promoters with an upstream gene oriented divergently. *Head-to-tail (H-T):* oriented in tandem. (B) Normalized H2A.Z, H2A, and input nucleosome tag coverage around the repetitive *RDN1* locus. A *red arrow* marks the TSS of the Pol I-controlled *RDN37-1* promoter. (C) Compiled tag coverage of 275 tRNA genes with and without rapamycin treatment. (D) Heat maps of input nucleosomes around the center of tRNA genes. (E) Nucleosome profiles of
*Figure 6 continued on next page*

*Figure 6 continued*

under-digested chromatin from the *TBP-FRB* strain centered at the dyad of the +1 nucleosomes. *Cyan* triangles mark the peak of fragile nucleosomes. (F) Same as (E) except the profiles were aligned at the dyad of fragile nucleosomes (*Kubik et al., 2015*).

The following source data and figure supplement are available for figure 6:

**Source data 1.** Nucleosome tag profiles around the +1 dyads for the under-digested *TBP-FRB* chromatin sample in (E).
**Source data 2.** A list of -1 nucleosomes that are flanked by two +1 nucleosomes.
**Figure supplement 1.** Compiled nucleosome tag coverage of 44 divergent promoters in *TBP-FRB* with a *bona fide* -1 nucleosome flanked by +1 nucleosomes.

whether these fragile nucleosomes contain H2A.Z (*Pradhan et al., 2015*). In some of our qChIP-seq experiments, where the chromatin was under-digested, fragile nucleosomes were observed (*Figure 6E*, *cyan arrowhead*). Strikingly, fragile nucleosomes are completely devoid of H2A.Z. The phenomenon is more obvious when sequencing tags around the dyads of 1,953 fragile nucleosomes were compiled (*Figure 6F*) (*Kubik et al., 2015*). When H2A.Z eviction is blocked by TBP depletion, no H2A.Z accumulation was observed at the fragile nucleosome positions (*Figures 6E–F*). This indicates H2A.Z deposition does not occur at these sites. The lower signal in input of the + RAP sample over the fragile nucleosome was due to slight over-digestion. Overall, our data suggest that H2A.Z deposition and eviction are highly localized at the +1 nucleosomes of Pol II-transcribed genes.

## H2A.Z accumulation in the absence of TBP can be used to determine internal and cryptic transcription start sites

The finding that H2A.Z dynamics revealed by the *TBP-FRB* mutant is linked to Pol II transcription start sites, raises the possibility that the Δ(Z/A) parameter can be used to identify cryptic, alternative, or previously unassigned transcription start sites. Indeed, the Δ(Z/A) parameter correctly marked the start sites of many previously annotated cryptic unstable transcripts (CUTs) (*Figure 7A*, *red arrows*, *CUT531* and *CUT445)* and noncoding RNAs (*Figure 7A*, *cyan arrows*, *SUT013* and *ICR1*) (*Bumgarner et al., 2012*; *Xu et al., 2009*). To identify novel start sites, the Δ(Z/A) profiles around the dyads of all 61,568 annotated nucleosomes of the yeast genome (*Kubik et al., 2015*; *Jiang and Pugh, 2009*) were sorted by *k*-means clustering (*Figure 7—figure supplement 1A*). Since the +1 nucleosome is expected to have a maximum of Δ(Z/A) signal centered around the dyad as opposed to a +2 or -1 nucleosome, which should have an off-centered Δ(Z/A) maximum contributed by the +1 neighbor, nucleosomes with symmetrically distributed Δ(Z/A) signal were selected (*Figure 7—figure supplement 1A*). The process was reiterated again with the remaining nucleosomes (n = 56,193) to identify +1 nucleosomes with weaker Δ(Z/A) signals (*Figure 7—figure supplement 1A*). This approach identified 4,576 potential +1 nucleosomes, of which 3,684 were known +1 nucleosomes (n = 6,427) of protein-coding and noncoding genes (*Jiang and Pugh, 2009*; *Kubik et al., 2015*; *Rhee et al., 2014*) (*Figure 7—figure supplement 1A*). The remaining 892 were not previously identified as +1 and therefore could represent novel +1 nucleosomes (*Figure 7—figure supplement 1B* and *Figure 7—source data 1*). CUTs are normally degraded by the RNA surveillance machinery but accumulate in the exosome subunit mutant *rrp6Δ* (*Xu et al., 2009*). Many of the new +1 nucleosomes are associated with subtle, unannotated CUTs revealed by the *rrp6Δ* strain (*Figure 7B*, *grey bars*). Therefore global comparison of H2A depletion and H2A.Z accumulation in the absence of the PIC provides a new parameter for transcription start site determination.

The Δ(Z/A) parameter also identifies alternative and novel start sites masked by other transcripts (*Figure 7C*). The transcriptional start site of *GDH2*, which encodes the NAD-linked glutamate dehydrogenase, has previously been mapped to a position on chromosome IV around 74,109 bp (*Figure 7C*, *left, green arrow*) (*Miller and Magasanik, 1991*; *Xu et al., 2009*). The identification of a new +1 nucleosome within the coding region of *GDH2* implies that there is a second start site ~1 kb downstream of the first (*Figure 7C*, *left, blue arrow*). Concordantly, microarray mRNA profiling data of wild-type cells (*Figure 7C*, *black dots*) showed an elevated mRNA level after the second start site indicating that a population of transcripts was indeed initiated from the internal start site (*Xu et al.,*

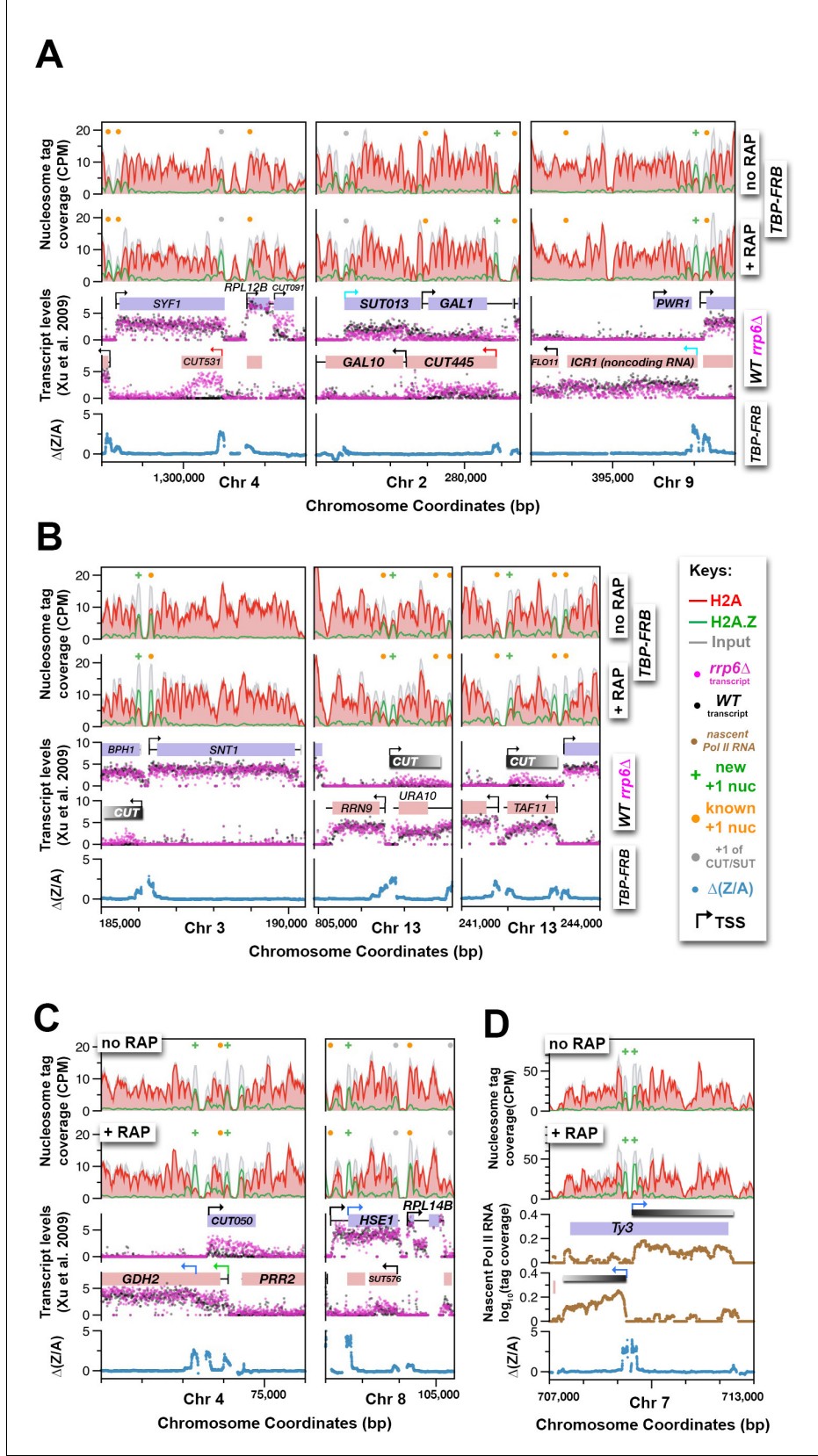

**Figure 7.** Using H2A.Z dynamics before and after TBP depletion to identify cryptic and alternative transcription start sites. (**A**) Genomic regions highlighting representative CUTs and noncoding RNAs, e.g. *CUT531, SUT013, CUT445, and ICR1*. Transcript data of wild-type (*black*) and *rrp6Δ* (*purple*)

*Figure 7 continued on next page*

*Figure 7 continued*

strains are from (*Xu et al., 2009*). (**B**) H2A.Z dynamics around subtle cryptic transcription start sites. *Gray bars*: unannotated CUTs. (**C**) Start sites masked by upstream transcripts were revealed by strong H2A.Z dynamics (*blue arrows*). (**D**) The cryptic divergent promoters within the coding region of a Ty3 element. *Brown traces:* Nascent Pol II RNA tag coverage. *Orange dots*: previously annotated +1 nucleosomes of protein coding genes (*Rhee et al., 2014*; *Kubik et al., 2015*). *Gray dots:* +1 nucleosomes of CUTs and SUTs. *Green '+' signs*: new +1 nucleosomes.

The following source data and figure supplement are available for figure 7:

**Source data 1.** Novel and previously annotated +1 nucleosomes identified by PIC-dependent H2A.Z dynamics.
**Figure supplement 1.** Identification of novel +1 nucleosomes using the Δ(Z/A) parameter associated with TBP depletion.

*2009*). In the case of *HSE1*, the novel start site identified by H2A.Z accumulation is likely the *bona fide* start site masked by an upstream SUT (*Figure 7C*, *right, blue arrow*). Further high resolution RNA-seq analysis of nascent Pol II transcripts showed that the originally annotated *HSE1* transcribed region (*Xu et al., 2009*) consists of two transcripts organized in close tandem (*Figure 7—figure supplement 1C*). Therefore, the originally annotated TSS of *HSE1* belongs to the upstream SUT that terminates immediately before the newly identified TSS of *HSE1* (blue arrow), although read-through transcripts emanating from the upstream TSS cannot be excluded (*Figure 7—figure supplement 1C*).

In another example, H2A.Z-enriched nucleosomes that flank an NDR within the Ty3 retrotransposon accumulated H2A.Z (and lost H2A) after TBP-FRB depletion, implying the presence of a pair of divergent promoters (*Figure 7D*). Indeed, our nascent Pol II RNA data, as well as standard RNA-seq analysis of mRNA, confirmed the presence of transcripts initiated from these sites (*Figure 7D* and *Figure 7—figure supplement 1D*). Cryptic antisense transcription has been reported for the Ty1 retrotransposon and the antisense Ty1 CUT is important for gene silencing and suppression of Ty1 mobility (*Berretta et al., 2008*). Therefore, the antisense Ty3 CUT identified here may play a similar role (*Figure 7D*, *gray bar*).

## Discussion

### Constitutive histone dynamics at +1 nucleosomes

The promoter-proximal NDR serves not only as a platform for the assembly of the transcription machinery, but also as a hub for the recruitment of an array of chromatin remodeling factors (*Rhee and Pugh, 2012*; *Venters et al., 2011*). For any given promoter, these proteins apparently do not co-exist but transiently bind and dissociate in a regulated fashion to set up a chromatin environment in and around the NDR that promotes an accurate transcriptional response. The challenge is to understand how these factors are recruited and dislodged, how they regulate the dynamics of chromatin structure, and in what order these steps occur. This work demonstrates that rapid, constitutive turnover of H2A.Z occurs genome-wide, even at infrequently transcribing genes. At the nucleosomes covering or immediately downstream of the start sites of Pol II transcripts, the relative enrichment of H2A.Z is a steady state maintained by two major opposing forces imposed by the SWR1 complex and the transcription machinery. SWR1 converts H2A nucleosomes to the H2A.Z-containing forms (*Figure 8*, step I) and the transcription machinery then actively disassembles them.

Given that the SWR1 complex is recruited to the promoter in part through its affinity for long, exposed linker DNA (*Ranjan et al., 2013*), preventing PIC assembly by TBP or Pol II removal could, in theory, allow more SWR1 to bind the NDR leading to H2A.Z accumulation. As for the infrequently transcribed genes, one could argue that the NDRs of these promoters might be occupied with partial PICs that may become accessible to SWR1 binding when TBP or Pol II is removed. Our data argue against this hypothesis in that Swr1 occupancy decreased, rather than increased, in response to TBP depletion. Therefore, it is the SWR1 complex already residing at the NDR before TBP depletion that is responsible for the increased H2A.Z deposition. However, since the Swr1 occupancy data are based on two promoters (with reliable Swr1 ChIP signal), aberrant SWR1 recruitment elsewhere in the genome cannot be ruled out.

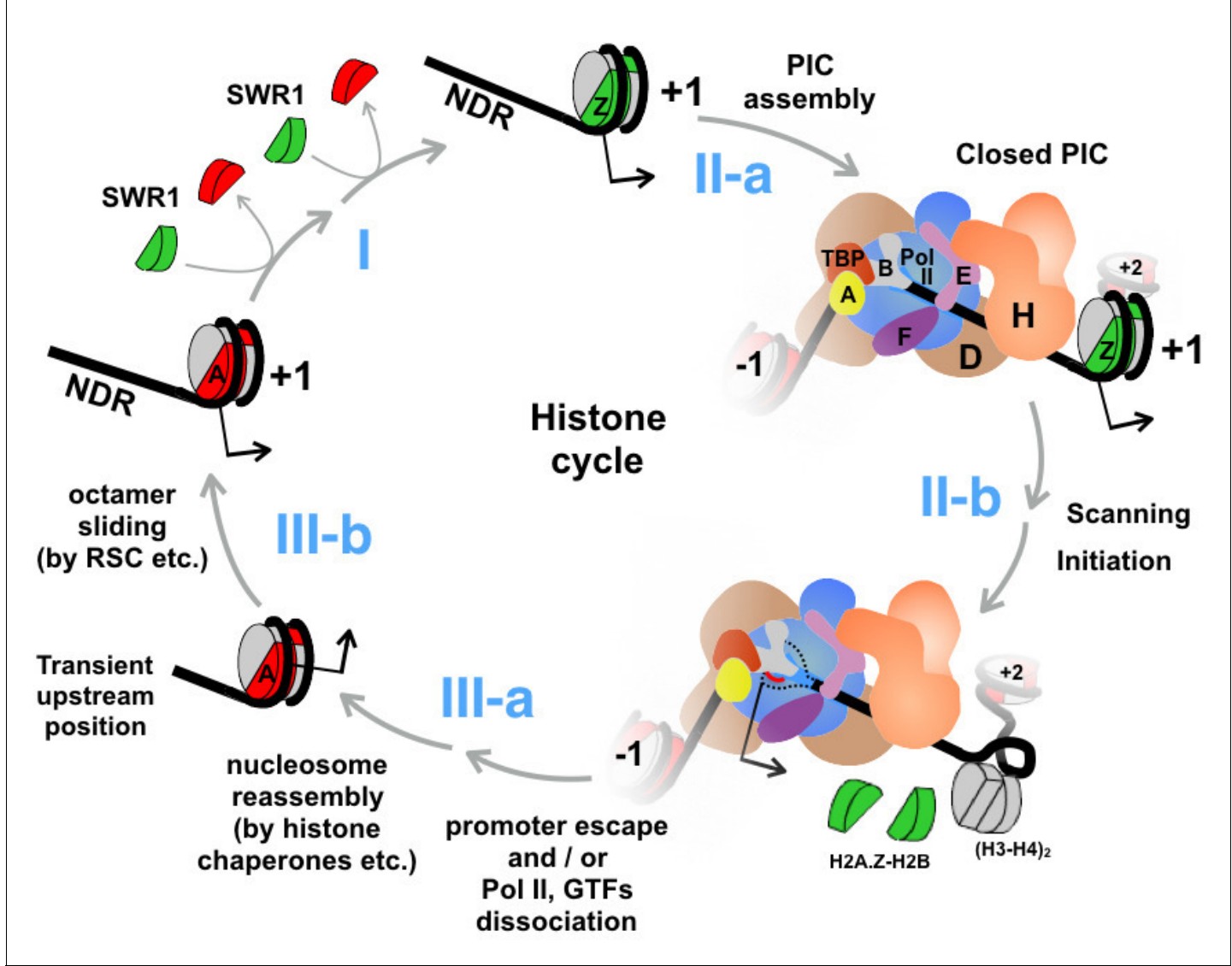

**Figure 8.** An updated histone cycle model. Step I: The SWR1 complex is recruited to the NDR and replaces the two nucleosomal H2A-H2B dimers with H2A.Z-H2B dimers. The H2A.Z-H2B dimers are always bound by histone chaperones *in vivo* but is omitted here for simplicity (*Luk et al., 2007*). Step II-a: Assembly of the PIC at NDR adjacent to the +1 H2A.Z nucleosome. General transcription factors are identified with single letters for simplicity. The model of PIC was adapted from (*Sainsbury et al., 2015*) Step II-b: The PIC engages the H2A.Z nucleosome. *Dotted black lines* depict the unstructured ssDNA, which is not drawn to scale. *Red line* depicts the nascent RNA. *Black arrow* indicates the transcription start site (TSS). $(H3-H4)_2$ tetramers are in *gray*, H2A.Z-H2B dimers in *green*, and H2A-H2B dimers in *red*. Step III-a: After the dissociation of Pol II and GTFs from the promoter, a canonical H2A nucleosome reassembles over the TSS but at an imprecise position that is likely too upstream of the TSS. Step III-b: Chromatin remodelers, such as RSC, slide the octamer downstream to the stereotypic +1 position completing the histone cycle.

The observation that Swr1 occupancy decreased upon TBP depletion is interesting as it most likely reflects the lower affinity of SWR1 for the H2A.Z nucleosome relative to the H2A nucleosome (*Ranjan et al., 2015*). As such, after SWR1 has generated the H2A.Z nucleosome product, the enzyme dissociates from the NDR. Alternatively but not exclusively, since optimal SWR1 recruitment requires H3 and H4 tail acetylation, inhibition of transcription could decrease histone acetylation, and thus indirectly impair SWR1 recruitment (*Liu et al., 2005*; *Raisner et al., 2005*; *Ranjan et al., 2013*).

## How does the transcription machinery engage the +1 H2A.Z nucleosome?

Perturbation of PIC assembly causes promoter-specific H2A.Z, but not H2A, to accumulate, suggesting that the constitutive disassembly occurs preferentially with H2A.Z-containing nucleosomes (*Figure 8*, steps II-a and II-b). Currently, it is not known whether the heterotypic AZ or the homotypic ZZ form is the preferred nucleosomal state for disassembly as our experiments did not distinguish between these forms. One explanation for the preference of H2A.Z eviction is that the PIC preferentially assembles on promoters that have a +1 H2A.Z nucleosome. Consistent with this idea, optimal TBP recruitment during oleate induction requires H2A.Z and the C-terminal domain of H2A.Z has been reported to bind RNA Pol II (*Adam et al., 2001*; *Wan et al., 2009*). Another nonexclusive possibility is that H2A.Z-containing nucleosomes are preferentially recognized by the PIC for disassembly. As such, when the PIC is fully engaged with the +1 H2A.Z nucleosome, some component(s) of the PIC actively disassembles the nucleosome, thereby allowing rapid, robust transcriptional activation. In support of this idea, rapid induction of *GAL1-10, PHO5* and the heat-responsive genes requires H2A.Z and is accompanied by preferential eviction of H2A.Z over H2A (*Santisteban et al., 2000*; *Venters et al., 2011*; *Zhang et al., 2005*). But regardless of the basis of how the preference for H2A.Z removal is achieved, our data suggest that the eviction of H2A.Z is, at least in part, coupled to some steps during transcription, rather than an independent process upstream of PIC assembly (*Figure 8*, step II-b).

A recent report that utilized ChIP and qPCR to monitor histone H3 occupancy showed that inactivation of transcription by the temperature sensitive alleles *rpb1-1* and *tbp ts-1*, led to promoter-proximal H3 accumulation, suggesting that the PIC is required to maintain an open chromatin state at the +1 position that is depleted of all histones (*Ansari et al., 2014*). In the current study, the input nucleosomal levels at the +1 positions were largely unchanged for most genes upon TBP (or Rpb1) depletion. This discrepancy may be an artifact caused by a normalization step in our study that equalizes the input tag counts of all samples. Therefore, if inhibition of transcription had indeed caused a global increase in nucleosomal levels, the input nucleosome of the TBP depleted sample would be undercounted. But because the qChIP-seq approach reports H2A.Z IP efficiency, even if the input nucleosome occupancy were indeed higher in the TBP depleted sample, the absolute increase of H2A.Z would in fact be greater in response to TBP depletion, further supporting our conclusion. Furthermore, the TBP depleted sample does not appear to be severely undercounted, as the least transcribed genes (bottom 3%) exhibited similar input nucleosome densities after normalization (*Figure 1—figure supplement 10*). Interestingly, another group has previously used MNase-seq to probe the change of chromatin organization in the *rpb1-1* mutant and showed that the +1 nucleosomes also did not accumulate when transcription was blocked (*Weiner et al., 2010*). We speculate that PIC-dependent disassembly of the +1 H2A.Z nucleosomes involves some metastable sub-nucleosomal intermediates (*Ramachandran et al., 2015*; *Rhee et al., 2014*). Since the DNA associated with the sub-nucleosomal species is likely hypersensitive to MNase digestion, it would be underrepresented in our input fraction. By contrast, standard H3 ChIP is likely less biased against the sub-nucleosomal species and therefore revealed H3 accumulation when transcription was blocked (*Ansari et al., 2014*).

Earlier electron microscopy and genome-wide mapping studies place the PIC immediately upstream of the TSS on the DNA, suggesting that the +1 H2A.Z nucleosome and the PIC (in the closed conformation) can coexist in many promoters (*Murakami et al., 2013*; *Rhee and Pugh, 2012*). It is currently unknown which step of the transcription process causes H2A.Z eviction. However, we speculate that promoter scanning mediated by TFIIH is the driving force (*Figure 8*, step II-b). A recent single-molecule study showed that yeast PIC repeatedly scans the promoter over an extended region (85 base pairs on average), leading to cycles of transcription bubble formation and collapse before committing to promoter escape (*Fazal et al., 2015*). Formation of such extended transcription bubbles in the NDR will overwind the DNA downstream, leading to a propensity to form positive supercoils in the +1 nucleosomal region. Given that DNA wraps the histone octamer in a left-handed turn (i.e. negative writhe) and that the promoter-distal end of the nucleosome is restricted to rotate freely, positive supercoiling will likely destabilize the +1 nucleosome. Repeated cycles of futile scanning without promoter escape may also occur at infrequently transcribing genes explaining the constitutive H2A.Z dynamics at those promoters. Although we favor a model in which

H2A.Z eviction is a direct consequence of the initiation process, we cannot exclude the possibility that a subsequent step of transcription, such as promoter escape or elongation, is the driving force as TBP depletion blocks all steps of transcription. We also cannot exclude TBP or the integrity of the PIC may be required to recruit another factor that functions to disassemble H2A.Z nucleosomes.

Since robust PIC-dependent H2A.Z eviction occurs even at promoters that infrequently fire, this provides an opportunity to use H2A.Z dynamics as a parameter to locate initiation sites that were previously missed by the conventional transcript mapping approach (*Xu et al., 2009*). Sites with strong H2A.Z dynamics were able to identify not only initiation sites of protein coding genes, SUTs and CUTs, but also sites that are masked by upstream transcripts. Although this approach cannot pinpoint the location of a TSS at base pair resolution, the sites are generally restricted within the identified +1 nucleosome near the edge adjacent to the NDR. Importantly, since PIC-dependent H2A.Z eviction is linked to an NDR, the direction of the predicted transcripts can be inferred based on the location of the NDR and is almost always pointing away from the NDR. Another limitation of this approach is that it may not identify the TSSs of promoters that are associated with low endogenous H2A.Z deposition activity and/or have low PIC-dependent H2A.Z eviction activity as in the case of the SAGA-dominating promoters.

## The *in vivo* contribution of the INO80 remodeler on H2A.Z eviction

Whether the INO80 complex plays a role in evicting H2A.Z at promoters has been controversial. Evidence supporting this idea comes from *in vitro* experiments showing that the INO80 complex can mediate a reaction that replaces the nucleosomal H2A.Z-H2B dimer with H2A-H2B, which supposedly opposes the reaction catalyzed by SWR1 (*Papamichos-Chronakis et al., 2011*), as well as *in vivo* experiments showing a global increase of H2A.Z occupancy in *ino80Δ* or *arp5Δ* (both components of the INO80 complex) strains (*Papamichos-Chronakis et al., 2006*; *Yen et al., 2013*). However, a more recent study showed *ino80Δ* mutants exhibited normal H2A.Z occupancy at promoters (*Jeronimo et al., 2015*). Our experiments agree with the latter study arguing that INO80 does not contribute to global eviction of H2A.Z, as no accumulation of H2A.Z was observed when Ino80 was depleted. One explanation for the discrepancy between these various experiments is that *ino80Δ* and *arp5Δ* are both associated with genome instability, resulting in aneuploidy that complicates the interpretation of genomic data (*Chambers et al., 2012* and Ashby Morrison personal communication). To circumvent this problem, we assayed H2A.Z occupancy within two hours of conditionally depleting Ino80 from the nucleus, thereby minimizing aneuploidy. Furthermore, the previous studies used ChIP-chip and ChIP-exo to measure H2A.Z occupancy (*Jeronimo et al., 2015*; *Papamichos-Chronakis et al., 2011*; *Yen et al., 2013*). However, these approaches may not be ideal for comparing H2A.Z occupancy in different strains. Standard microarray normalization involves scaling with data averages that could mask any global occupancy change (*Quackenbush, 2002*). ChIP-exo measures the distribution of crosslinking points between the immunoprecipitated protein and DNA that could be influenced by variation of crosslinking efficiency in mutants with chromatin dynamics defects (*Yen et al., 2013*). The qChIP-seq approach conducted here takes into account the input, flow-through, and IP fractions of each IP reaction and uses the immuno-depleted signal in the flow-through to scale the IP signal. Therefore, occupancy data reflects IP efficiency relative to input at individual nucleosomal sites and is not influenced by global occupancy change. It is also less sensitive to variation in crosslinking efficiency as formaldehyde crosslinking serves only to preserve the histone-DNA interaction during the pull down. Our data showed that Ino80 depletion did not cause H2A.Z accumulation, arguing against the INO80 remodeler being the main evictor of H2A.Z in the constitutive turnover process. This observation however is not inconsistent with INO80 being recruited to remodel specific promoters in response to induction as seen in the *PHO5* and *GAL* genes (*Barbaric et al., 2007*; *Rosonina et al., 2014*).

Although INO80 is not involved in the constitutive eviction of H2A.Z at the +1 positions, our data suggest that INO80 does contribute to the positioning of nucleosomes near the promoters. While the sliding defect associated with Ino80 depletion is reproducible and consistent with previous studies (*Yao et al., 2016*; *Yen et al., 2012*), the defect is small compared to that associated with mutants in genes encoding other chromatin remodelers, such as *RSC*, *CHD1*, *ISW1*, and *ISW2* (*Gkikopoulos et al., 2011*; *Hartley and Madhani, 2009*; *Ocampo et al., 2016*; *Whitehouse et al., 2007*). Therefore, the role of INO80 in nucleosome positioning could be indirect or redundant with another remodeling factor.

## An updated model of histone dynamics at yeast promoters

We propose an updated 'histone cycle' model to unify the molecular events that lead to the promoter-specific H2A.Z dynamics (*Figure 8*). First, ATP-dependent remodelers, such as RSC, are recruited to the promoter region by sequence specific DNA-binding factors to establish the NDR (*Badis et al., 2008*; *Hartley and Madhani, 2009*) (*Figure 8*, step III-b). SWR1 is recruited to the NDR by its affinity for extended stretches of naked DNA and, to lesser extent, promoter-specific histone acetylation (*Hartley and Madhani, 2009*; *Ranjan et al., 2013*). SWR1 then converts the canonical AA nucleosome to the AZ and ZZ chromatin states sequentially (*Luk et al., 2010*) (*Figure 8*, step I). For promoters that contain a fragile nucleosome within the NDR region (omitted in *Figure 8* for simplicity), this structure is somehow refractory to the SWR1-mediated histone replacement reaction. SWR1 dissociates from the promoters after depositing H2A.Z at the +1 position due to its lower affinity for the H2A.Z nucleosomal product (*Ranjan et al., 2015*). The PIC assembles at the NDR adjacent to the newly formed +1 H2A.Z nucleosome (*Rhee and Pugh, 2012*). At some point when the PIC engages the TSS, the H2A.Z nucleosome disassembles. This may involves sub-nucleosomal intermediates before all histones are completely evicted from the DNA. Following promoter escape or unproductive dissociation of the PIC, a nucleosome with the major histones, H2A, H2B, H3, and H4, reassembles completing the histone cycle (*Figure 8*, step III-a).

The promoter-specific histone dynamics described by the histone cycle appears to be a general phenomenon in yeast. We show that the Pol II transcription machinery has a major contribution to the eviction step of H2A.Z in the histone cycle. Together, chromatin remodelers and the transcription machinery choreograph the movement of histones within the promoter leading to a dynamic chromatin architecture conducive for transcription. How these enzymes are coordinated to set the histone cycle in motion will be the next important question.

## Materials and methods

### Yeast strains and culture conditions

The genotypes of the yeast strains used in this study are listed in *Supplementary file 1*. The anchor-away strains HHY221, HHY170 and HHY209 were obtained from *Euroscarf* (*Haruki et al., 2008*). The H2A.Z FLAG tagged strains yEL152 (no *FRB* tag control), yEL154 (*SPT15-FRB-GFP*), yEL170 (*RPB1-FRB*), and yEL189 (*INO80-FRB-GFP*) were constructed by integrating a *2xFLAG-URA3* fragment at the 3' end of the *HTZ1* gene in the precursor strains HHY221, yEL098, yEL090, and yEL123, respectively. yEL098 is a tetrad segregant of the diploid HHY221 x HHY209 strain and yEL090 is a segregant of the diploid HHY221 x HHY170. The *2xFLAG-URA3* fragment was obtained by PCR amplification of the *pRS416-HTZ1-2xFLAG* plasmid described in ref. (*Mizuguchi et al., 2004*). Successful integration was confirmed by colony PCR and sequencing.

The *SWC5-FRB* (yEL219) and *SWC5-FRB TBP-FRB* (yEL220) strains were generated by integrating the *FRB-HIS3MX6* fragment at the 3' end of the *SWC5* coding region in yEL152 and yEL154, respectively, using the one-step gene replacement procedure (*Longtine et al., 1998*). The *FRB-HIS3MX6* fragment was amplified by PCR using the plasmid *pFA6a-FRB-HIS3MX6* (P30579, *Euroscarf*) as template (*Haruki et al., 2008*). Similarly, the *INO80-FRB-GFP* precursor strain (yEL123) was generated by transforming HHY221 (yEL044) with a *FRB-GFP-HIS3MX6* PCR product targeting the 3' end of *INO80*. The *FRB-GFP-HIS3MX6* fragment was amplified from the plasmid *pFA6a-FRB-GFP-HIS3MX6* (P30581, *Euroscarf*) (*Haruki et al., 2008*).

For microscopy, yeast cells were grown in YPD at 30°C to an $OD_{600}$ of 1 before rapamycin was added to a final concentration of 1 μg/mL. Cells before and after the addition of rapamycin were removed at the indicate times and were fixed with 4% formaldehyde for 5 min followed by washing with 1x PBS and staining with 1 μg/mL DAPI. For the microscopy analysis of the *INO80-FRB-GFP* cells in *Figure 4—figure supplement 1C*, live cells were used in order to shorten the lag time between imaging and fixation before the qChIP-seq analysis.

The yeast cells used in quantitative ChIP were cultured at 30°C in 2 L of yeast extract-peptone-dextrose (YPD) to an optical density 600 ($OD_{600}$) of 0.8 before 1 μg/mL rapamycin was added and incubated for the indicated duration. The rapamycin-treated culture, as well as the untreated control, were fixed with 3% paraformaldehyde for 7.5 min and quenched by 0.12 M glycine as previously

described (*Luk et al., 2010*). Cell pellets equivalent to 400 mL culture were aliquoted and washed with 1x PBS before flash frozen and stored at –80°C.

## qChIP-seq

Cells equivalent to 400-mL culture volume were spheroplasted and lysed with a 7-mL dounce homogenizer (piston B, *Wheaton, Millville, NJ*) as previously described (*Luk et al., 2010*). After centrifuging the lysate (~1000 µL) at 13,000 x g for 10 min at 4°C, the chromatin-enriched pellet was washed 3 times with buffer A [50 mM HEPES (pH 7.5), 80 mM NaCl, 0.25% Triton X-100, 2x protease inhibitor cocktail (cOmplete EDTA-free, *Roche, Switzerland*)]. The pellet was resuspended in 1/2 x lysate volume of buffer A plus 1 mM of $CaCl_2$. The suspension was pre-incubated to 37°C for 2 min before digestion with 1.5–1.8 U/µL MNase (*Worthington, Lakewood, NJ*) at 37°C for 10–20 min. The concentration of MNase and duration of digestion were empirically determined for each sample to achieve ~90% mono-nucleosomal form. After stopping the digestion reaction by the addition of 10 mM EGTA, the sample was centrifuged at 20,400 x g for 15 min at 4°C. The supernatant, which contained the soluble nucleosomes, was cleared by passage through a low-binding PVDF filter (Ultrafree GV, *EMD Millipore, Billerica, MA*).

Before the soluble nucleosomes were subjected to quantitative ChIP, a 50 µL aliquot was saved. The DNA extracted from this sample represents the input nucleosome fraction. The remaining sample (~500 µL) was diluted 10-fold in buffer B [25 mM HEPES-KOH pH 7.6, 80 mM KCl, 1 mM EDTA, 1x protease inhibitors (PI, which consists of 1.7 mg/mL PMSF, 3.3 mg/mL benzamidine hydrochloride, 0.1 mg/mL pepstatin A, 0.1 mg/mL leupeptin, 0.1 mg/mL chymostatin)] followed by incubation with 1/20x volume (~250 µL resin volume) of anti-FLAG M2 affinity gel (A2220, *Sigma-Aldrich, St. Louis, MO*) at 4°C for 4 hr. The flow-through of the IP reaction would represent the H2A nucleosome fraction. The bead-bound H2A.Z nucleosomes were washed three times with buffer C (25 mM HEPES-KOH pH 7.6, 0.3 M KCl, 1 mM EDTA, 0.01% NP-40, 1x PI) and eluted with 2x resin volume (~500 µL) of buffer D (25 mM HEPES-KOH pH 7.6, 0.3 M KCl, 1 mM EDTA, 0.1% NP-40, 0.1% DOC, 1x PI, 0.5 µg/µL 3xFLAG peptides) at 4°C overnight. The eluate was cleared by passage through a GV 0.22 µm filter (*EMD Millipore*). The FLAG eluate would represent the H2A.Z nucleosome fraction.

To extract the nucleosomal DNA, the sample was adjusted to 400 µL with molecular grade water followed by the addition of 32 µL 5 M NaCl, 8 µL 0.5 M EDTA, 20 µL 10% SDS, 2.5 µL 20 µg/µL glycogen (*Thermo Fisher, Waltham, MA*), and 8 µL of 20 mg/mL proteinase K (*Thermo Fisher*). The mixture was incubated at 55°C for one hour to facilitate protein digestion and 65°C for > 15 hr to reverse the crosslinks. The DNA was purified by standard phenol-chloroform extraction followed by ethanol-NaOAc precipitation. The pellet was resuspended in 100 µL of 5 µg/mL RNase (*Roche*) in 1x TE pH 8.0 and incubated at 37°C for > 1 hr. The resulting nucleosomal DNA was purified using the QIAquick spin column (*Qiagen, Germany*) and quantified by the Qubit HS assay (*Thermo Fisher*).

To prepare the DNA libraries for sequencing, 30 ng of nucleosomal DNA was treated with 5 U of alkaline phosphatase (CIP, *NEB, Ipswich, MA*) for 1 hr at 37°C followed by the addition of 20 µg glycogen (*Thermo Fisher*). The mixture was then purified by standard phenol-chloroform extraction and ethanol-NaOAc precipitation. The DNA was resuspended in 12 µL of the TruSeq Resuspension Buffer (*Illumina, San Diego, CA*) and quantified by the Qubit HS assay. Ten nanograms of CIP-treated DNA was applied to the TruSeq ChIP workflow (*Illumina*) with the following modifications. After the end-repair step, instead of using the AMPure beads for purification, the sample was purified by standard phenol-chloroform extraction and ethanol-NaOAc precipitation. The adapter-ligated DNA was amplified on a PCR machine by 15 thermal cycles. The PCR product was quantified by the Qubit assay (*Thermo Fisher*) and the quality was verified by agarose electrophoresis and SYBR green staining (*Thermo Fisher*). Densitometry was performed on a Typhoon FLA9500 scanner station installed with the ImageQuant TL software (*GE Healthcare, Pittsburgh, PA*). Paired-end sequencing (33 cycles) was performed on a MiSeq sequencer (*Illuimna*).

## Bioinformatics

Sequencing reads were aligned to the *S. cerevisiae* genome (SGD version R64-1-1) using Bowtie 1.1.2 (*Langmead et al., 2009*). Aligned data of the H2A.Z, H2A and input nucleosome fractions were processed without smoothing using a combination of custom Python scripts and BEDTools

programs (*Quinlan, 2014*). They were plotted along the genome either as tag coverage (density covered by paired-end reads) or tag counts (density of mid-points of paired-end reads). The amplitude of the H2A.Z (Z), H2A (A), and input (T or total) nucleosome profiles was scaled using an approach called 'TAZ normalization' (*Figure 1—figure supplement 4*). First, the tag counts within 44 reference regions (called the no-Z-zones), which are enriched for H2A but depleted for H2A.Z (*Figure 1—figure supplement 4A*, *Figure 1—source data 1*), were determined by Bowtie. The normalization factor $m$, representing the ratio of no-Z-zones tag count of the input fraction over that of the H2A fraction, was used to scale the H2A nucleosome profile (*Figure 1—figure supplement 4B*, *Figure 1—source data 2*). To normalize the H2A.Z nucleosome profile for each IP reaction (technical replicate), the tag counts around 4,738 promoters were compiled around the +1 dyads for the H2A, H2A.Z and input fractions. The normalization factor $n$, was determined by a curve fitting algorithm such that the sum of the compiled profiles of H2A and H2A.Z ($m \times$ H2A + $n \times$ H2A.Z) equals, to a first approximation, the compiled profile of the input (*Figure 1—figure supplement 4B*). The Python script used to determine the $m$ and $n$ values can be found in *Source code 1*. Normalization accuracy was verified using a visualization tool, which runs on Datagraph (*Visual Data Tools Inc.*) (*Supplementary file 3*). A more detailed procedure on how to use the Python script and the Datagraph program can be found in *Supplementary file 4*. Finally, the input profiles for each IP reaction were converted to counts per million (CPM) by the normalization factors $c$. The products of $c \times m$ and $c \times n$ represent the final normalization factors for the H2A and H2A.Z nucleosome profiles, respectively. The compiled nucleosomal profiles for each qChIP-seq experiment can be found in *Figure 1—source data 3*, *4* and *5*, *Figure 3—source data 1*, *Figure 4—source data 1* and *Figure 6—source data 1*, and the normalization factors in *Figure 1—source data 2*.

All qChIP-seq experiments with the exception of the experiment in *Figure 3* represent the average of two biological replicates (i.e from independent cultures). In *Figure 3*, the qChIP-seq data of *SWC5-FRB* and *SWC5-FRB TBP-FRB* strains represent the averages of two independent IP reactions (technical replicates) of the same biological sample. The biological replicate for *SWC5-FRB* is shown in *Figure 3—figure supplement 2*, which included additional time points after rapamycin treatment. Data averaging was performed on the normalized tag coverage or tag counts after the TAZ normalization step; therefore, sequencing reads were not pooled.

## Immunoblotting and ChIP-qPCR analyses

Antibodies against the yeast Swr1 and H2A.Z were gifts of Carl Wu. The anti-FLAG (F1804) and the anti-H2A (39235) antibodies were purchased from *Sigma-Aldrich* and *Active Motif (Carlsbad, CA)*, respectively. For H2A and H2A.Z western, both antibodies were used at a dilution of 1:2,000. ChIP-qPCR analysis was performed as previously described with the following modifications (*Aparicio et al., 2005*). ChIP reactions were performed using 1.25 mg dynabeads conjugated to Protein A or G. Five microliter of anti-Swr1 was used in each ChIP reaction. qPCR was performed on a LightCycler 96 Real-Time PCR system (*Roche*). The primers used in qPCR are listed in *Supplementary file 2*.

## Nascent Pol II transcript sequencing

Nascent Pol II transcript sequeuncing was performed essentially according to (*Churchman and Weissman, 2012*) but with the following modifications. Briefly, yeast cells expressing *RPB3-3xFLAG* (yEL297) were cultured in 6 L YPD to a final OD of 2. Washed cells were pulverized in Buffer E (20 mM HEPES-KOH pH 7.6, 20% Glycerol, 50 mM KOAc, 1 mM EDTA, 1x PI) for 15 min using the Freezer Mill (*SPEX, Metuchen, NJ*). The resulting whole cell extract was incubated with 40 U/mL of DNase I (DPRF, *Worthington*), 25 Units/mL of SUPERase RNase Inhibitor (*Thermo Fisher*) and 10 mM MnCl$_2$ at 4°C for 30 min before centrifugation at 48,000 x g for 30 min. Immunopurification was performed using 12 mL (1-L culture equivalent) of cleared lysate and 400 µL (bed-volume) of anti-FLAG agarose (*Sigma-Aldrich*) on a rotator at 4°C for 2.5 hr. After washing with Buffer F (20 mM HEPES-KOH pH 7.6, 10% glycerol, 110 mM KOAc, 0.1% NP-40, 1x PI), elution was performed with 400 µL of 0.5 µg/µL 3xFLAG peptide in Buffer G (20 mM HEPES-KOH pH 7.6, 10% glycerol, 110 mM KOAc, 0.1% NP-40, 1 mM DTT, 1x PI) on a rotator at 4°C for 2 hr. RNA co-eluted with the Rpb3-3xFLAG Pol II complex was purified on a miRNeasy column (*Qiagen*) according to the manufacturer protocol. To further enrich for the nascent Pol II RNA, ribosomal RNA was depleted using the Yeast Ribo-Zero

kit (*Illumina*). cDNA library was prepared using the TruSeq stranded mRNA library prep kit (*Illumina*). Sequencing reads were aligned to the yeast genome with Bowtie as described above. A combination of custom Python scripts and BEDTools programs (*Quinlan, 2014*) were used to generate the tag coverage data for *Figure 7D* and *Figure 7—figure supplement 1C*. The Pol II nascent RNA data represents the average of two biological replicates.

## Data access

Sequencing data are available at the National Center for Biotechnology Information Sequence Read Archive (http://www.ncbi.nlm.nih.gov/sra) under accession number SRP051897 and are also accessible through BioProject ID PRJNA271808. Processed genome-wide qChIP-seq data are available through Dryad under the Digital Object Identifier: doi:10.5061/dryad.dj782 (*Tramantano et al., 2016*).

## Acknowledgement

We thank Nancy Hollingsworth for critical reading of the manuscript, Carl Wu and members of the Wu lab for helpful discussions and sharing of reagents, our reviewers for insightful critiques, Aaron Neiman and David Matus for assistance in fluorescence microscopy, Ashby Morrison for discussion of unpublished data, and Alisa Yurovsky and Bruce Futcher for sequencing and bioinformatics support.

## Additional information

### Funding

| Funder | Grant reference number | Author |
|---|---|---|
| National Institute of General Medical Sciences | RO1 GM104111 | Ed Luk |
| National Institute of General Medical Sciences | T32 GM008468 | Michael Tramantano |

The funders had no role in study design, data collection and interpretation, or the decision to submit the work for publication.

### Author contributions

MT, LS, EL, Conception and design, Acquisition of data, Analysis and interpretation of data, Drafting or revising the article; CA, CS, Acquisition of data, Analysis and interpretation of data, Contributed unpublished essential data or reagents; DL, ZL, MC, Acquisition of data, Analysis and interpretation of data

### Author ORCIDs

Ed Luk, http://orcid.org/0000-0002-6619-2258

## Additional files

### Supplementary files

• Supplementary file 1. Table of yeast strains used in this study.

• Supplementary file 2. Table of primers used in ChIP-qPCR analyses.

• Supplementary file 3. The visualization tool used to verify the normalization process.

• Supplementary file 4. Detailed instructions on how to use the TAZ normalization script and the visualization tool.

• Supplementary file 5. An example of the INFILE of the python script corresponding to the raw data of a TBP-FRB + RAP replicate.

• Supplementary file 6. An example of the INFILE of the python script corresponding to the raw data of a TBP-FRB no RAP replicate.

• Source code 1. The Python script used to perform TAZ normalization.

## Major datasets

The following datasets were generated:

| Author(s) | Year | Dataset title | Dataset URL | Database, license, and accessibility information |
|---|---|---|---|---|
| Tramantano M, Sun L, Au C, Labuz D, Liu Z, Chou M, Shen C, Luk E | 2015 | Constitutive turnover of histone H2A.Z at yeast promoters requires the preinitiation complex | http://www.ncbi.nlm.nih.gov/bioproject/?term=PRJNA271808 | Publicly available at the NCBI BioProject database (accession no: PRJNA271808) |
| Tramantano M, Sun L, Au C, Labuz D, Liu Z, Chou M, Shen C, Luk E | 2016 | Constitutive turnover of histone H2A.Z at yeast promoters requires the preinitiation complex | http://www.ncbi.nlm.nih.gov/sra/?term=SRP051897 | Publicly available at the NCBI Short Read Archive (accession no: SRP051897) |
| Tramantano M, Sun L, Au C, Labuz D, Liu Z, Chou M, Shen C, Luk E | 2016 | Data from: Constitutive turnover of histone H2A.Z at yeast promoters requires the preinitiation complex | http://dx.doi.org/10.5061/dryad.dj782 | Available at Dryad Digital Repository under a CC0 Public Domain Dedication |

The following previously published datasets were used:

| Author(s) | Year | Dataset title | Dataset URL | Database, license, and accessibility information |
|---|---|---|---|---|
| Rhee HS, Bataille AR, Zhang L, Pugh BF | 2014 | Subnucleosomal Structures and Nucleosome Asymmetry across a Genome | http://www.ncbi.nlm.nih.gov/sra/?term=SRA059355 | Publicly available at the NCBI Short Read Archive (accession no: SRA059355) |
| Kubik S, Bruzzone MJ, Jacquet P, Falcone JL, Rougemont J, Shore D | 2015 | Two Distinct Promoter Nucleosome Architectures at Protein-Coding Genes in Yeast | http://www.ncbi.nlm.nih.gov/geo/query/acc.cgi?acc=GSE73337 | Publicly available at the NCBI Gene Expression Omnibus (accession no: GSE73337) |
| Lipson D, Raz T, Kieu A, Jones DR, Giladi E, Thayer E, Thompson JF, Letovsky S, Milos P, Causey M | 2009 | Accurate Quantitation of the Yeast Transcriptome by Single Molecule Sequencing | http://www.ncbi.nlm.nih.gov/sra/?term=SRA008810 | Publicly available at the NCBI Short Read Archive (accession no: SRA008810) |
| Rhee HS, Pugh BF | 2012 | Genome-wide structure and organization of eukaryotic pre-initiation complexes | http://www.ncbi.nlm.nih.gov/sra/?term=SRA046523 | Publicly available at the NCBI Short Read Archive (accession no: SRA046523) |
| Xu Z, Wei W, Gagneur J, Perocchi F, Clauder-Munster S, Camblong J, Guffanti E, Stutz F, Huber W, Steinmetz LM | 2009 | Transcription profiling of wild type yeast grown with ethanol, glucose and galactose and the deletion mutant of Rrp6 to identify transcription start and end positions | https://www.ebi.ac.uk/arrayexpress/experiments/E-TABM-590/ | E-TABM-590 |

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
