## [Decision Letter]

[Editors’ note: this article was originally rejected after discussions between the reviewers, but the authors were invited to resubmit after an appeal against the decision.]

Thank you for submitting your work entitled "Constitutive H2A.Z turnover at yeast promoters requires the preinitiation complex" for consideration by *eLife*. Your article has been reviewed by three peer reviewers, one of whom, Julie Ahringer, is a member of our Board of Reviewing Editors and the evaluation has been overseen by Jessica Tyler as the Senior Editor. Our decision has been reached after consultation between the reviewers. Based on these discussions and the individual reviews below, we regret to inform you that your work will not be considered further for publication in *eLife*.

The reviewers appreciated that you were addressing an interesting question, but agreed that your data did not support your central conclusion that the transcription machinery has a direct role in H2A.Z disassembly (explained in detail in the individual comments). For example, the multiple effects of Kin28 depletion meant it was not possible to assign phenotypes to a particular pathway, so the role of promoter escape could not be addressed. A significant amount of new work would be needed to make the paper acceptable, which is not compatible with a revision in *eLife*.

Reviewer #1:

This paper investigates the mechanism of eviction of +1 H2A.Z nucleosomes at promoters. The authors deplete factors that are expected to affect PIC assembly, H2A.Z deposition, or transcription, and then assess for H2A.Z abundance using a semi-quantitative ChIP method. There are some interesting initial findings here but at present they do not allow clear and strong conclusions. Some conclusions drawn are stronger than the data allow, sometimes due to the nature of the perturbation and sometimes due to issues with data analyses.

Explanation of how replicates were treated is not sufficient and there is no assessment of reproducibility of results. These need to be explicitly explained in the methods. The only details I could find were in the Figure 1 legend: “The data in A-D represent the unsmoothed mean of 3 to 5 independent ChIP reactions (technical replicates) from two separate cultures (biological replicates).” The methods should indicate how many replicates were done for each experiment, the concordance between replicates, and how pooling of data was done. The authors need to show that similar results were obtained in independent experiments.

It is not appropriate to use the change of Z/A ratio to measure H2A.Z dynamics. The numerator and denominator are not independent, so linear changes in occupancy result in non-linear changes in the factor. For example, if Z changes from 90% to 80%, (10% change), then the change in Z/A ratio would be 90/10 compared to 80/20. 9/4 = 2.25 fold. Also, this measure is biased, as it is differently sensitive at different levels of H2A.Z occupancy. If the change H2A.Z changed from 50% to 40%, the change in Z/A ratio would only be 1.5, whereas it becomes sensitive again at low H2A.Z levels. This also causes problems in the statistical treatment since Z/A ratio values vary significantly for a similar change in H2A.Z. Instead, the change in H2A.Z should be directly assessed and compared, as the authors eventually do: "We also plotted the change of H2A.Z-over-input, Δ(H2A.Z/input), and found this parameter less sensitive but more unbiased (Figure 1—figure supplement 5)." The ratio assays should be removed.

The experiment simultaneously depleting TBP and Swc5 is not interpretable, as the result will depend on the relative depletion of the two factors, which is unknown and not controllable. The individual depletions are however, informative.

The Kin28 depletion to assess the impact of promoter escape is problematic because Kin28 loss both reduces the amount of PIC assembly and prevents escape when a PIC is assembled. As the authors don't know the level of PIC assembly defect compared to the promoter escape defect, the experiment is not interpretable. A ChIP of TBP could be helpful here.

Reviewer #2:

This paper makes the interesting observation that depletion of TBP from the nucleus, which precludes assembly of the PIC, results in an increase in H2A.Z at the +1 nucleosome. This persuasively implies an antagonism, and subsequent experiments suggest that the increase in H2A.Z is due to decreased removal rather than increased recruitment (but see below). This isn't too surprising, but it's a nice result. Less convincing is an experiment arguing that depletion of the Kin28 subunit of TFIIH, which supposedly allows PIC assembly but not promoter escape, does not produce the same effect on H2A.Z. The authors want to conclude that some post-PIC, pre-escape step actively targets H2A.Z nucleosomes at +1, but here I found that their models went too far into unsupported speculation (see below).

Major issues:

1) In the third paragraph of the Introduction section, the authors state that it's completely unknown what dissociates H2A.Z. I thought perhaps they were unaware of the paper from Craig Peterson claiming it's Ino80 (Papamichos-Chronakis et al., 2011), but then they cite this paper in the sixth paragraph starting of the Introduction section. Even if they don't find the Papamichos-Chronakis convincing, in my opinion their discussion is overly biased. Is it fair to characterize the Papamichos-Chronakis data for occurring under "certain in vitro conditions", when it's not that different from the Swr1C data? The authors also imply that the Ino80 model lacks in vivo support. It's fine to cite the Jeronimo paper as showing no effect of Ino80 on Htz1 ChIP, but you have to acknowledge that ChIPs in the earlier paper from Papamichos disagree.

2) Sixth paragraph, Introduction section. On a related point, I don't see how H3/H4 turnover argues against specific remodelers taking out H2A.Z. There's no reason complete nucleosome eviction (by Swi/Snf, for example) is mutually exclusive with additional specific removal of the H2A/H2B or H2A.Z/H2B dimers.

3) Figure 2 is not completely convincing. While the compiled tag counts on the left suggest the double depletion is less severe than Swc5 alone, the scatter plots at the right make it clear the change is pretty subtle. It's hard to reconcile the numbers from the left panel (sixth paragraph of subsection “PIC assembly is required for genome-wide H2A.Z eviction at promoters from both active and quiescent genes”: 7.5 fold loss of H2A.Z, versus 2.4 fold) with what looks like a more modest difference in the scatter plot. In any case, even with the more severe numbers, the results seem to me to strongly argue that there are probably multiple mechanisms related to clearing H2A.Z, only one of which is transcription.

4) Subsection “H2A.Z eviction occurs at distinct stages of transcription”. The experiment using Kin28 depletion as a method for blocking promoter escape just isn't convincing, especially since this also affects PIC assembly (as noted by the authors). How can we be sure that the difference in H2A.Z response between Kin28-FRB and TBP-FRB isn't due to differences in the kinetics or completeness of nuclear depletion? In the paper from the Robert lab cited for the Kin28 depletion, they actually rested most of their conclusions on chemical inhibition of the kinase rather than the anchor away system. That would be a better way to test the hypothesis in this paper.

5) Subsection “H2A.Z accumulation in the absence of TBP can be used to determine internal and cryptic transcription start sites”. The data for the internal start site in GHD2 looks good, and there is also the CUT in the antisense direction, so it's nice to see the double peak in the Htz1 signal representing internal bidirectional transcription. However, I just don't see the alleged internal start site in *HSE1*. Also, it's probably worth noting in the text that using H2A.Z as a marker for internal start sites will not tell you which direction transcription is going.

6) First paragraph Discussion section. Perhaps I missed it, but what data shows constitutive turnover of H2A.Z at quiescent genes? Please cite which figure. How is quiescence even defined in yeast, where even "silent" genes may fire occasionally? Did the authors look at truly silenced promoters, for example, those within HMR and HML?

7) Second paragraph Discussion section. There's no justification for invoking recognition of Htz1 by the PIC. It could just be that Htz1 is put at ends of NFRs preferentially, and firing off of the PIC dissociates it. Indeed, the model proposed by the authors where TFIIH-mediated scanning displaces H2A.Z does not require PIC recognition or even direct interaction with +1 nucleosome.

8) Same section as above. Similarly, I can see no basis whatsoever in the data to say that H2A disassembly is transcription independent. I would say authors' model should instead predict that TBP depletion should result in an overall increase in +1 occupancy, independent of whether there's H2A.Z is there or not. Would the authors' normalization scheme obscure overall changes in nucleosome occupancy?

Reviewer #3:

This manuscript investigates the effects of pre-initiation complex (PIC) recruitment on promoter nucleosome dynamics. An elegant conditional depletion approach is used to achieve this. The main observation made is that enrichment of the histone variant H2A.Z increases upon removal of the PIC. This indicates that the PIC normally is involved in a process that depletes H2A.Z from +1 nucleosomes countering the action of the Swr1 complex that normally acts to increase H2A.Z at these locations. This suggests that enrichment of H2A.Z at promoters is normally the result of a dynamic equilibrium between incorporation and removal. The manuscript then proceeds to show that sites of H2A.Z turnover can be used to identify cryptic start sites for transcription. These finding represent important new insights into how chromatin is organised at promoters. Overall the data are of high technical quality and presented clearly.

The interpretation made in the manuscript is that the pre-initiation complex directly acts to displace H2A.Z from promoter nucleosomes. For example the possibility that migration of a bubble of separated DNA at the PIC could drive this is discussed. However, there appear to be alternate explanations that are not mentioned. One is that TBP is required for recruitment of or is co-recruited with another factor that acts to destabilise H2A.Z containing nucleosomes. A second is that an unidentified factor acts with partial redundancy to incorporate H2A.Z, and that this factor is excluded from, promoters when the PIC is present. The authors should consider incorporating discussion of the potential for alternate explanations that do not involve the action of the PIC directly removing H2A.Z.

It would also appear to be quite easy for the authors to look into whether the effects on H2A.Z are more strongly correlated with SAGA or TFIID regulated genes. It would be useful if this could be mentioned in the discussion.

[Editors’ note: what now follows is the decision letter after the authors submitted for further consideration.]

Thank you for submitting your article "Constitutive H2A.Z turnover at yeast promoters requires the preinitiation complex" for consideration by *eLife*. Your article has been reviewed by three peer reviewers, one of whom, Julie Ahringer, is a member of our Board of Reviewing Editors, and another is Tom Owen-Hughes (Reviewer #3). The evaluation has been overseen by Jessica Tyler as the Senior Editor.

The reviewers have discussed the reviews with one another and the Reviewing Editor has drafted this decision to help you prepare a revised submission.

Summary:

The manuscript provides an important step forward in identifying that engagement of the PIC contributes to H2A.Z cycling at promoters. It clearly illustrates that the steady state levels observed are the result of an equilibrium between directed incorporation and removal. This is likely to be of significant general interest. They also provide evidence that INO80 does not play a role in +1 nucleosome dynamics. This helps to address a topical and somewhat controversial subject. The results advance our understanding of promoter regulation.

Essential revisions:

1) The authors propose a model where the PIC somehow actively removes Htz1, even though there's no precedent for such an enzymatic activity. Depletion of TBP or Pol II doesn't simply reduce PIC assembly; it blocks all subsequent steps, most notably transcription itself. The data in this paper are also consistent with promoter escape, elongation, or termination being involved in removal of Htz1. Statements concluding an active role of the PIC in removal of Htz1 should be toned down, e.g.:

"Impact statement: The transcription machinery disassembles the promoter-proximal H2AZ nucleosome […]";

"These findings suggest that the Pol II transcription machinery plays a more active role than previously through in the remodeling of chromatin structure within promoters";

"We show that the Pol II transcription machinery has a chromatin remodeling activity…".

2) As their model figure shows, this is a cycle. Given the documented affinity of SwrC for the free DNA in the NDR, a much simpler idea is that binding of the PIC blocks SwrC association. Depletion of TBP or Pol II would allow more Swr binding, leading to an increase in Htz1 occupancy. Even on infrequently transcribed genes, there may still be PIC assembly without productive transcription. And Swr1 may continually recycle Htz1 even on non-transcribed genes with an unoccupied NDR. These alternative possible mechanisms for changes in Htz1 occupancy should be discussed.

3) A paper from Randy Morse's lab (Ansari et al. (2014) Mediator, TATA-binding Protein, and RNA Polymerase II Contribute to Low Histone Occupancy at Active Gene. JBC 289, 14981-14995) used TBP depletion or Kin28 inhibition and concluded that the entire +1 nucleosome (rather than only H2AZ) decreased in a "PIC-dependent" manner. This paper should be cited and the differences discussed.

4) Yen et al. 2013 observed that in an Arp5 mutant H2A.Z occupancy is significantly increased at promoters. This paper is not cited. It should be cited and included in the discussion of studies that have looked at the role of Ino80 in H2A.Z turnover. Currently 2 studies provide support for INO80 acting in H2A.Z removal and 2 against. The value of the manuscript would be improved if a plausible explanation for these discrepancies could be identified.

5) The relative profiles of H2A and H2A.Z in no RAP in Figure 1 (TBP-FRB and no FRB backgrounds) differ, with lower relative H2A.Z in the no FRB background. The Z/A ratio is higher in TBP-FRB with no RAP than in untagged control with no RAP. This difference is also evident in the correlation diagrams in Figure 1—figure supplement 5, comparing the two no RAP experiments. Can the authors comment on these differences and how they might affect confidence in the observed changes in +RAP conditions? How do these differences relate to the variability in Z/A ratio between replicates? The difference is important, because the authors used the untagged control to determine significantly affected regions in Figure 1 (explained in Figure 1—figure supplement 6).

6) To confirm the qChIP-seq results, the authors tested H2A.Z levels at three nucleosomal regions and two coding regions by qPCR. To increase confidence in this control, the FT fraction should also be assessed, along with levels of H2A in both fractions.

---

## [Author Response]

[Editors’ note: the author responses to the first round of peer review follow.]

*Reviewer #1:*

This paper investigates the mechanism of eviction of +1 H2A.Z nucleosomes at promoters. The authors deplete factors that are expected to affect PIC assembly, H2A.Z deposition, or transcription, and then assess for H2A.Z abundance using a semi-quantitative ChIP method. There are some interesting initial findings here but at present they do not allow clear and strong conclusions. Some conclusions drawn are stronger than the data allow, sometimes due to the nature of the perturbation and sometimes due to issues with data analyses.

Overall, we have improved the analysis of the qChIP-seq data by replacing the nonlinear (H2A.Z/H2A) parameter with the (H2A.Z/input) parameter for several key figures to allow more quantitative comparison. We strengthen the conclusion about the role of the PIC in H2A.Z eviction by the inclusion of Pol II subunit Rpb1, another component of the PIC. We agree that the effect of the Kin28 perturbation is difficult to interpret; therefore, the Kin28 depletion experiment is now removed. Instead we added new data showing that INO80, a remodeler previously reported to displace nucleosomal H2A.Z, has insignificant role in H2A.Z eviction in vivo. This new work provides a new perspective on what contribution the different chromatin remodeling pathways has on H2A.Z eviction.

Explanation of how replicates were treated is not sufficient and there is no assessment of reproducibility of results. These need to be explicitly explained in the methods. The only details I could find were in the Figure 1 legend: The data in A-D represent the unsmoothed mean of 3 to 5 independent ChIP reactions (technical replicates) from two separate cultures (biological replicates). The methods should indicate how many replicates were done for each experiment, the concordance between replicates, and how pooling of data was done. The authors need to show that similar results were obtained in independent experiments.

We now indicate in the Methods section the number of replicates done for each experiment. The definitions of technical and biological replicates and the procedure of data averaging are also stated in the Methods. Analysis for the concordance between replicates can be found in the Figure 1—figure supplement 5, Figure 1—figure supplement 6 and Figure 4—figure supplement 1.

It is not appropriate to use the change of Z/A ratio to measure H2A.Z dynamics. The numerator and denominator are not independent, so linear changes in occupancy result in non-linear changes in the factor. For example, if Z changes from 90% to 80%, (10% change), then the change in Z/A ratio would be 90/10 compared to 80/20. 9/4 = 2.25 fold. Also, this measure is biased, as it is differently sensitive at different levels of H2A.Z occupancy. If the change H2A.Z changed from 50% to 40%, the change in Z/A ratio would only be 1.5, whereas it becomes sensitive again at low H2A.Z levels. This also causes problems in the statistical treatment since Z/A ratio values vary significantly for a similar change in H2A.Z. Instead, the change in H2A.Z should be directly assessed and compared, as the authors eventually do: "We also plotted the change of H2A.Z-over-input, Δ(H2A.Z/input), and found this parameter less sensitive but more unbiased (Figure 1—figure supplement 5)." The ratio assays should be removed.

Reviewer #1 correctly pointed out that the change in Z/A ratio is non-linear and is therefore not appropriate to measure H2A.Z dynamics. In fact, we did not do justice to our own data by using the Δ(Z/A) parameter as it disproportionally highlights sites that exhibit an increase in relative H2A.Z and underemphasizes those that exhibit a decrease. This became a problem when we compared how much less H2A.Z is in the Swc5 depletion mutant compared to the Swc5 TBP double depletion mutant. We have now replaced Δ(Z/A) with the (H2A.Z/input) parameter for comparison between different samples. We kept Δ(Z/A) for highlighting sites with strong H2A.Z dynamics and for identifying the +1 nucleosome of novel transcription start sites.

It should be pointed out that Δ(Z/A) is the *difference* between (H2A.Z/H2A)_[+RAP]_ and (H2A.Z/H2A)_[noRAP]_, not the ratio between the two as illustrated in the example of the Reviewer’s comment above. Similarly the change in (H2A.Z/input) represents the difference between (H2A.Z/input)_[+RAP]_ and (H2A.Z/input)_[noRAP]_.

The experiment simultaneously depleting TBP and Swc5 is not interpretable, as the result will depend on the relative depletion of the two factors, which is unknown and not controllable.. The individual depletions are however, informative.

We failed to point out that we have previously performed a time course of Swc5 depletion and found that 30 min of rapamycin treatment caused H2A.Z to deplete to a pre-baseline level (see new Figure 3—figure supplement 1). Therefore, the change in H2A.Z occupancy in the Swc5 single depletion is a rough estimate of H2A.Z decay rate. In theory, if PIC assembly is not involved in H2A.Z eviction, the rate of H2A.Z decay caused by Swc5 depletion should not be affected by TBP depletion. On the contrary, if the PIC is the only H2A.Z evictor *and* if the anchor-away depletion rates of TBP and Swc5 are the same, H2A.Z deposition and eviction should cancel out; therefore the steady-state H2A.Z levels should be unchanged before and after Swc5 and TBP codepletion. What we observed is that H2A.Z is depleted but to a lesser extent in the Swc5 and TBP co-depletion than in the single Swc5 depletion. This could mean that Swc5 anchor-away depletion is faster than TBP depletion and thus the deposition of H2A.Z is shut off earlier than the eviction of H2A.Z. Another explanation is that TBP and Swc5 are depleted at similar rates but the PIC is not the only H2A.Z evictor. Both of these possibilities are still in agreement with PIC being a contributor to H2A.Z eviction.

The Kin28 depletion to assess the impact of promoter escape is problematic because Kin28 loss both reduces the amount of PIC assembly and prevents escape when a PIC is assembled. As the authors don't know the level of PIC assembly defect compared to the promoter escape defect, the experiment is not interpretable. A ChIP of TBP could be helpful here.

We agree. The Kin28 data are now removed.

*Reviewer #2:*

This paper makes the interesting observation that depletion of TBP from the nucleus, which precludes assembly of the PIC, results in an increase in H2A.Z at the +1 nucleosome. This persuasively implies an antagonism, and subsequent experiments suggest that the increase in H2A.Z is due to decreased removal rather than increased recruitment (but see below). This isn't too surprising, but it's a nice result. Less convincing is an experiment arguing that depletion of the Kin28 subunit of TFIIH, which supposedly allows PIC assembly but not promoter escape, does not produce the same effect on H2A.Z. The authors want to conclude that some post-PIC, pre-escape step actively targets H2A.Z nucleosomes at +1, but here I found that their models went too far into unsupported speculation (see below).

We agree that the Kin28 data needs more work. Therefore, we have now removed this data from the new submission. Instead, we added new data showing the consequence of relative H2A.Z occupancy upon INO80 depletion. We directly compare the contribution of PIC-dependent and INO80-dependent H2A.Z eviction. We observed that PIC is a major contributor of H2A.Z eviction, but not INO80.

*Major issues:*

*1) In the third paragraph of the Introduction section, the authors state that it's completely unknown what dissociates H2A.Z. I thought perhaps they were unaware of the paper from Craig Peterson claiming it's Ino80 (Papamichos-Chronakis et al., 2011), but then they cite this paper in the sixth paragraph starting of the Introduction section. Even if they don't find the Papamichos-Chronakis convincing, in my opinion their discussion is overly biased. Is it fair to characterize the Papamichos-Chronakis data for occurring under "certain* in vitro *conditions", when it's not that different from the Swr1C data? The authors also imply that the Ino80 model lacks in vivosupport. It's fine to cite the Jeronimo paper as showingno effect of Ino80 on Htz1 ChIP, but you have to acknowledge that ChIPs in the earlier paper from Papamichos disagree.*

We thank Reviewer #2 for pointing out this problem in our previous manuscript. Our views were biased by our own unpublished data showing that Ino80 depletion has no effect on genome wide H2A.Z eviction. We have added these INO80 data to the current manuscript and explained why we think INO80 is not the main evictor of H2A.Z in vivo.

2) Sixth paragraph of the Introduction section. On a related point, I don't see how H3/H4 turnover argues against specific remodelers taking out H2A.Z. There's no reason complete nucleosome eviction (by Swi/Snf, for example) is mutually exclusive with additional specific removal of the H2A/H2B or H2A.Z/H2B dimers.

Point taken. The argument is now removed.

3) Figure 2 is not completely convincing. While the compiled tag counts on the left suggest the double depletion is less severe than Swc5 alone, the scatter plots at the right make it clear the change is pretty subtle. It's hard to reconcile the numbers from the left panel (sixth paragraph of subsection “PIC assembly is required for genome-wide H2A.Z eviction at promoters from both active and quiescent genes”: 7.5 fold loss of H2A.Z, versus 2.4 fold) with what looks like a more modest difference in the scatter plot. In any case, even with the more severe numbers, the results seem to me to strongly argue that there are probably multiple mechanisms related to clearing H2A.Z, only one of which is transcription.

As mentioned in our response to Reviewer #1 comment, the Δ(Z/A) parameter underemphasized the difference between the single Swc5 depletion and the Swc5 TBP co-depletion. We have re-plotted the data using the change in (H2A.Z/input) before and after rapamycin, and the difference is now more obvious (Figure 3). To further illustrate our point, we present the average (H2A.Z/input) values of genes grouped and sorted by transcription frequency before and after rapamycin treatment. The decay of H2A.Z occupancy after Swc5 depletion is clearly delayed by TBP depletion. We also discuss the possibility of a PIC-independent mechanism(s) contributing to the eviction of H2A.Z. However, this PIC-independent mechanism is unlikely mediated by INO80 as mentioned above (Results section).

4) Subsection “H2A.Z eviction occurs at distinct stages of transcription”. The experiment using Kin28 depletion as a method for blocking promoter escape just isn't convincing, especially since this also affects PIC assembly (as noted by the authors). How can we be sure that the difference in H2A.Z response between Kin28-FRB and TBP-FRB isn't due to differences in the kinetics or completeness of nuclear depletion? In the paper from the Robert lab cited for the Kin28 depletion, they actually rested most of their conclusions on chemical inhibition of the kinase rather than the anchor away system. That would be a better way to test the hypothesis in this paper.

Point taken. The *kin28-as* experiment is a good suggestion. But as mentioned above, the experiment related to Kin28 is now removed and we plan to continue to work on this as a separate study.

*5) Subsection “H2A.Z accumulation in the absence of TBP can be used to determine internal and cryptic transcription start sites”. The data for the internal start site in GHD2 looks good, and there is also the CUT in the antisense direction, so it's nice to see the double peak in the Htz1 signal representing internal bidirectional transcription. However, I just don't see the alleged internal start site in HSE1.*

It was misleading to say that the new +1 H2A.Z nucleosome within the annotated *HSE1* transcript marks an alternative TSS. In fact, the new +1 H2A.Z peak marks the bona fide TSS of *HSE1* that is masked by an upstream stable unannotated transcript (SUT) (Figure 7). Since the transcript data [Xu Z. et al. Nature. (2009)] used to plot Figure 7 are from microarray, the resolution may be too low to see the termination of the SUT and the initiation of *HSE1*. Therefore, we performed RNA-seq on Pol II-associating nascent RNA. Indeed, we were able to see a very short region between the SUT and the *HSE1* that is largely devoid of RNA, indicating that the SUT terminates immediately before the bona fide TSS of *HSE1* (blue arrow). Therefore, the previously annotated TSS of *HSE1* is incorrect as it belongs to the SUT, and the old *HSE1* transcribed region actually consists of two transcripts organized in close tandem. This figure is now included as Figure 7—figure supplement 1.

Also, it's probably worth noting in the text that using H2A.Z as a marker for internal start sites will not tell you which direction transcription is going.

In fact, when combining with the information of the relative position of the NDR, the H2A.Z marker can predict the direction of transcription. The increase of H2A.Z upon TBP depletion is almost always next to an NDR. The direction of the TSS is always pointing away from the NDR. This point is now indicated in the Discussion (subsection “How does the transcription machinery engage the +1 H2A.Z nucleosome?”).

6) First paragraph Discussion section. Perhaps I missed it, but what data shows constitutive turnover of H2A.Z at quiescent genes? Please cite which figure. How is quiescence even defined in yeast, where even "silent" genes may fire occasionally? Did the authors look at truly silenced promoters, for example, those within HMR and HML?

We agree that even the least expressed genes in yeast fire occasionally. Therefore, we now describe these genes as infrequently transcribing instead of quiescent. Figure 3 and Figure 3—figure supplement 1 of the revised manuscript show rapid turnover of H2A.Z occurs at both active and infrequently transcribing genes, and thus constitutive. The experiment showed that average H2A.Z occupancy was diminished to almost baseline (within 30 min) after Swc5 depletion even for genes with a transcriptional frequency <1 mRNA/hr. In addition, we showed that the depletion was slowed when TBP was simultaneously depleted, suggesting that the eviction is at least partially dependent on PIC activity.

7) Second paragraph Discussion section. There's no justification for invoking recognition of Htz1 by the PIC. It could just be that Htz1 is put at ends of NFRs preferentially, and firing off of the PIC dissociates it. Indeed, the model proposed by the authors where TFIIH-mediated scanning displaces H2A.Z does not require PIC recognition or even direct interaction with +1 nucleosome.

We agree that the proposed twist-induced nucleosome eviction should be nondiscriminative for H2A and H2A.Z nucleosome. But there is evidence suggesting that PIC assembly occurs preferentially on promoters with H2A.Z. For example, optimal recruitment of PIC (TBP) during gene induction requires H2A.Z [Wen Y. et al. MCB., (2009)]. It has also been reported that the C-terminus of H2A.Z interacts with RNA Pol II [Adam, M. et al. MCB. (2001)]. Given that the +1 nucleosome of any given promoter is alternating between the H2A and H2A.Z states in a cell population and that PIC assembly occurs preferentially on promoters with +1 H2A.Z nucleosome, PIC-mediated nucleosome disassembly will preferentially affect H2A.Z. Our data agree with this model as relative H2A.Z occupancy at +1 disproportionately increases upon PIC depletion. This point is now discussed in the text (second paragraph Discussion section).

8) Same section as above. Similarly, I can see no basis whatsoever in the data to say that H2A disassembly is transcription independent. I would say authors' model should instead predict that TBP depletion should result in an overall increase in +1 occupancy, independent of whether there's H2A.Z is there or not. Would the authors' normalization scheme obscure overall changes in nucleosome occupancy?

The comment in the original manuscript that says H2A disassembly is transcription independent is misleading and is now removed.

Given that transcription activity is known to evict histones, it is reasonable to expect that PIC depletion should cause an increase in overall nucleosome occupancy (input) around the +1 positions. For the same reason, the +1 nucleosome occupancy of the least active genes should remain largely unchanged upon PIC depletion. For the top 3% of most highly transcribed genes [sorted by Lipson D. et al. Nat. Biotech. (2009)], PIC depletion does exhibit an increase in nucleosome occupancy (read coverage) at the +1 positions (sum between -60 to +60 bp around +1 dyads) (Figure 1—figure supplement 10, red dots, red line = linear regression). However, the nucleosome occupancy for the bulk of the +1 positions (open circles, dotted line) has a similar trend as the bottom 3% least transcribed genes (blue dots and blue line) before and after PIC depletion, indicating that the nucleosome occupancy after PIC depletion is not dramatically different for most +1 nucleosomes.

Most importantly, we now present all H2A.Z data as relative H2A.Z occupancy or (H2A.Z/input), which represents H2A.Z occupancy divided by input nucleosome occupancy. For the most highly transcribing genes where (H2A.Z/input) is increased after PIC depletion, underestimation of nucleosome occupancy in the input of the PIC depleted sample means the absolute H2A.Z accumulation after PIC depletion is in fact even higher, further supporting our conclusion.

*Reviewer #3:*

*This manuscript investigates the effects of pre-initiation complex (PIC) recruitment on promoter nucleosome dynamics. An elegant conditional depletion approach is used to achieve this. The main observation made is that enrichment of the histone variant H2A.Z increases upon removal of the PIC. This indicates that the PIC normally is involved in a process that depletes H2A.Z from +1 nucleosomes countering the action of the Swr1 complex that normally acts to increase H2A.Z at these locations. This suggests that enrichment of H2A.Z at promoters is normally the result of a dynamic equilibrium between incorporation and removal. The manuscript then proceeds to show that sites of H2A.Z turnover can be used to identify cryptic start sites for transcription. These finding represent important new insights into how chromatin is organised at promoters. Overall the data are of high technical quality and presented clearly.*

*The interpretation made in the manuscript is that the pre-initiation complex directly acts to displace H2A.Z from promoter nucleosomes. For example the possibility that migration of a bubble of separated DNA at the PIC could drive this is discussed. However, there appear to be alternate explanations that are not mentioned. One is that TBP is required for recruitment of or is co-recruited with another factor that acts to destabilise H2A.Z containing nucleosomes.*

We now indicate this possibility in the Discussion.

A second is that an unidentified factor acts with partial redundancy to incorporate H2A.Z, and that this factor is excluded from, promoters when the PIC is present.

In the Swc5 depletion experiment where rapamycin-induced depletion was allowed for 60 minutes (Figure 3—figure supplement 1), we observed an almost complete depletion of H2A.Z occupancy. This indicates that SWR1 is likely the sole depositor of H2A.Z at the promoters of budding yeast. This point is now highlighted in the text (subsection “Constitutive PIC-dependent H2A.Z eviction is associated with promoters of active and infrequently transcribed genes that are generally TFIID enriched”).

The authors should consider incorporating discussion of the potential for alternate explanations that do not involve the action of the PIC directly removing H2A.Z.

As mentioned above, we have incorporated an alternative explanation for the removal of H2A.Z that does not involve the action of the PIC directly. Another controversial explanation in the literature is that the INO80 remodeler is involved in H2A.Z removal. As indicated in the introduction and discussion of the current text, Papamichos-Chronakis M., et al. Cell (2010) has reported that *ino80Δ* accumulated H2A.Z at promoters but a more recent study by Jeronimo C. et al., Mol. Cell (2015) did not observe this. One caveat with both of these studies is the use of *ino80Δ* deletion mutants, which are known to cause genetic instability and aneuploidy. We have added an experiment in which Ino80 was depleted by Anchor-away and then H2A.Z occupancy was measured soon afterward, thereby avoiding the genomic instability problem. Our results are consistent with Jeronimo C. et al. in that the Ino80 depletion has no effect on H2A.Z occupancy.

*It would also appear to be quite easy for the authors to look into whether the effects on H2A.Z are more strongly correlated with SAGA or TFIID regulated genes. It would be useful if this could be mentioned in the discussion.*

The correlation analysis of SAGA and TFIID against sites with strong H2A.Z dynamics is a good suggestion and is now shown as Figure 2 with interpretation described in the text.

[Editors’ note: the author responses to the re-review follow.]

*Essential revisions:*

*1) The authors propose a model where the PIC somehow actively removes Htz1, even though there's no precedent for such an enzymatic activity. Depletion of TBP or Pol II doesn't simply reduce PIC assembly; it blocks all subsequent steps, most notably transcription itself. The data in this paper are also consistent with promoter escape, elongation, or termination being involved in removal of Htz1.*

We have now discussed the possibility regarding the transcriptional steps other than initiation involved in H2A.Z eviction (subsection “How does the transcription machinery engage the +1 H2A.Z nucleosome?”).

Statements concluding an active role of the PIC in removal of Htz1 should be toned down, e.g.:

"Impact statement: The transcription machinery disassembles the promoter-proximal H2AZ nucleosome […]";

*"These findings suggest that the Pol II transcription machinery plays a more active role than previously through in the remodeling of chromatin structure within promoters";*

*"We show that the Pol II transcription machinery has a chromatin remodeling activity…".*

The impact statement and the statements regarding the role of PIC in H2A.Z removal were toned down in the revised text.

2) As their model figure shows, this is a cycle. Given the documented affinity of SwrC for the free DNA in the NDR, a much simpler idea is that binding of the PIC blocks SwrC association. Depletion of TBP or Pol II would allow more Swr binding, leading to an increase in Htz1 occupancy. Even on infrequently transcribed genes, there may still be PIC assembly without productive transcription. And Swr1 may continually recycle Htz1 even on non-transcribed genes with an unoccupied NDR. These alternative possible mechanisms for changes in Htz1 occupancy should be discussed.

A new paragraph (second paragraph in the Discussion) was added to discuss the possibility that increased SWR1 recruitment could be an explanation for the promoter-specific H2A.Z accumulation in response to TBP or Pol II depletion.

3) A paper from Randy Morse's lab (Ansari et al. (2014) Mediator, TATA-binding Protein, and RNA Polymerase II Contribute to Low Histone Occupancy at Active Gene. JBC 289, 14981-14995) used TBP depletion or Kin28 inhibition and concluded that the entire +1 nucleosome (rather than only H2AZ) decreased in a "PIC-dependent" manner. This paper should be cited and the differences discussed.

A new paragraph (second paragraph in subsection “How does the transcription machinery engage the +1 H2A.Z nucleosome”) dedicated to the discussion of the differences reported by Ansari et al. (2014) was added.

4) Yen et al. 2013 observed that in an Arp5 mutant H2A.Z occupancy is significantly increased at promoters. This paper is not cited. It should be cited and included in the discussion of studies that have looked at the role of Ino80 in H2A.Z turnover. Currently 2 studies provide support for INO80 acting in H2A.Z removal and 2 against. The value of the manuscript would be improved if a plausible explanation for these discrepancies could be identified.

We have now cited the paper by Yen et al. 2013. Plausible explanations for these discrepancies are given in the Discussion.

5) The relative profiles of H2A and H2A.Z in no RAP in Figure 1 (TBP-FRB and no FRB backgrounds) differ, with lower relative H2A.Z in the no FRB background. The Z/A ratio is higher in TBP-FRB with no RAP than in untagged control with no RAP. This difference is also evident in the correlation diagrams in Figure 1—figure supplement 5, comparing the two no RAP experiments. Can the authors comment on these differences and how they might affect confidence in the observed changes in +RAP conditions? How do these differences relate to the variability in Z/A ratio between replicates? The difference is important, because the authors used the untagged control to determine significantly affected regions in Figure 1 (explained in Figure 1—figure supplement 6).

The text was revised to discuss this issue (subsection “PIC assembly is required for genome-wide H2A.Z eviction at yeast promoters”). Please note the order of the supplemental figures in the revised manuscript is different.

As mentioned in the revised text, the endogenous H2A.Z level in the no RAP control of *TBP-FRB* is higher than that of the untagged control because the *TBP-FRB* allele is partially defective, thereby predisposing the *TBP-FRB* cells for H2A.Z accumulation.

The different anchor-away strains were compared based on the *change* in relative H2A.Z, which is represented by the difference in the (H2A.Z/input) ratios {i.e. (H2A.Z/input)_[RAP]_ – (H2A.Z/input)_[no RAP]_} (Figure 1, Figure 3, Figure 4, and Figure 1—figure supplement 6). This approach normalizes the relative H2A.Z at the ground state (no RAP), highlighting the functional consequence caused by the depletion of TBP or other factors in question. Therefore, any differences in the relative H2A.Z levels in the ground state should not affect the confidence in the observed changes in the +RAP conditions. It should be pointed out that the difference in the (Z/A) ratios was used only in the Figure 1—figure supplement 7 to illustrate that this parameter for detecting change in H2A.Z is more sensitive but non-linear.

Variability of (H2A.Z/input) between replicates should manifest as spread *perpendicular* to the diagonal axis (dotted black line) of Figure 1—figure supplement 5. However, any H2A.Z eviction defect will cause the data points to shift *along* the diagonal axis towards the upper right corner and thus should not affect the variability between replicates.

*6) To confirm the qChIP-seq results, the authors tested H2A.Z levels at three nucleosomal regions and two coding regions by qPCR. To increase confidence in this control, the FT fraction should also be assessed, along with levels of H2A in both fractions.*

The qPCR analysis of the FT fraction relative to the input at the three +1 nucleosomal positions and the two coding regions is now included as Figure 1—figure supplement 11. The results indicate a reciprocal decrease of H2A after TBP depletion consistent with the qChIP-seq data at the similar regions. For the +RAP and no RAP samples, the H2A levels in the input and the FT fractions were similar, indicating that the decrease of H2A is not due to non-specific binding of H2A nucleosomes to the anti-FLAG beads. This data can be found in Figure 1—figure supplement 11.